SciPost Physics

# A Global View of the EDM Landscape

Skyler Degenkolb[1], Nina Elmer[2],
Tanmoy Modak[2], Margarete Mühlleitner[3], and Tilman Plehn[2,4]

**1** Physikalisches Institut, Universität Heidelberg, Germany
**2** Institut für Theoretische Physik, Universität Heidelberg, Germany
**3** Institute for Theoretical Physics, Karlsruhe Institute of Technology, Karlsruhe, Germany
**4** Interdisciplinary Center for Scientific Computing (IWR), Universität Heidelberg, Germany

## Abstract

**Permanent electric dipole moments (EDMs) are sensitive probes of the symmetry structure of elementary particles, which in turn is closely tied to the baryon asymmetry in the universe. A meaningful interpretation framework for EDM measurements has to be based on effective quantum field theory. We interpret the measurements performed to date in terms of a hadronic-scale Lagrangian, using the SFitter global analysis framework. We find that part of this Lagrangian is constrained very well, while some of the parameters suffer from too few high-precision measurements. Theory uncertainties lead to weaker model constraints, but can be controlled within the global analysis.**

# 1   Introduction

While the Standard Model (SM) is structurally complete, it fails to explain key observations and therefore fails to qualify as a complete theory of elementary particles. The two leading short-comings are a missing dark matter agent and a missing explanation of the baryon asymmetry in the Universe. For the latter, the Sakharov conditions [1] tell us precisely which structures would be required: (i) C- and CP-violation, (ii) baryon number violation, and (iii) a deviation from thermal equilibrium. The first condition is especially interesting, because we can read it off the fundamental Lagrangian and build, for instance, models for electroweak baryogenesis around it [2–5]. In the SM, CP symmetry is violated through the fermion mixing among three generations and through the adjoint gluon field strength. Measurements of the neutron EDM, consistent with zero, clearly show that CP violation in QCD is too small to explain the observed baryon asymmetry [6–11]. Physics beyond the Standard Model (BSM), explaining the baryon asymmetry, should then violate CP to a greater extent and with observable consequences.

Over recent years, EDM measurements have been performed on a wide range of particles, atoms, and molecules [12–15]. None of them has been able to confirm a signal for CP violation. To judge their combined impact on BSM physics, we need to combine them in a consistent framework. Such a global analysis must start from a Lagrangian and express the experimental limits on all EDMs in terms of its fundamental parameters. One choice of Lagrangian is the hadronic-scale Lagrangian, describing the interactions of nucleons, pions, and electrons at the GeV scale [12, 15–17]. Alternatively, we can combine the EDM measurements at the weak scale, using an effective extension of the renormalizable SM Lagrangian [13, 17–22]. Going beyond effective field theories (EFTs), an ultimate link to the cosmological motivation requires a UV-complete extension of the SM at the weak scale, for instance leptoquark models [17], extended Higgs sectors [23–25], left-right symmetry [26], or supersymmetry [27–32]. Technically, all global analyses [33] face similar challenges which we will tackle in this paper for the hadronic-scale Lagrangian.

We perform a global EDM analysis with the SFITTER analysis tool [34–38], with its focus on the comprehensive treatment of uncertainties. Our analysis will provide state-of-the-art limits on the multi-dimensional model parameter space, and it will allow us to judge the impact of new or proposed measurements and to identify shortcomings in relating measurements to fundamental parameters[1], while remaining easy to repeat or adapt in response to new theoretical or experimental inputs. For this purpose it is crucial that we interpret all measurements in the same fundamental physics framework and include all uncertainties, including theory uncertainties, even though they lack a statistical interpretation [39].

We start by introducing the consistent hadronic-scale Lagrangian with properly chosen Wilson coefficients for our global analysis in Sec. 2. We then use this Lagrangian to provide predictions for the measured EDMs, as detailed in Sec. 3. We start our global analysis without theory uncertainties in Sec. 4, to understand the relations among different EDM measurements in terms of the hadronic-scale Lagrangian. To extract correlations and limits on single model parameters we employ a profile likelihood. We find that the current EDM measurements define a subspace of well-constrained model parameters and an orthogonal subspace of poorly constrained parameters with narrow correlation patterns. Adding theory uncertainties on the relations between Lagrangian parameters and observables in Sec. 4.5 degrades the interpretation in terms of fundamental physics. We emphasize that this degradation does not cut into the discovery potential of EDM measurements, as probes of fundamental symmetries of elementary particles, but it hampers their interpretation as limits in fundamental physics.

---

[1]Only a proper fundamental physics interpretation can make full use of experimental limits.

## 2 EDM Lagrangian

Because CP violation is motivated by cosmology and can be related to physics beyond the SM at and above the weak scale, we start by introducing it into the weak-scale Lagrangian in Sec. 2.1, relate it to the GeV-scale Lagrangian in Sec. 2.2, and use simple matching arguments to simplify this hadronic-scale Lagrangian which we use as the interpretation framework for our global analysis in Sec. 2.3. A detailed analysis of CP-violating new physics at the electroweak scale includes a renormalization group evolution from the GeV-scale to the electroweak scale and will not be part of this first study. Instead, we will focus here on a global analysis at the hadronic scale and the role of correlations and theory uncertainties.

### 2.1 Weak-scale Lagrangian

The operators generating CP violation within and beyond the SM-Lagrangian, neglecting CP violation in the neutrino sector [13], appear in the Lagrangian

$$\mathscr{L}_{\text{CPV}} = \mathscr{L}_{\text{CKM}} + \mathscr{L}_{\bar{\theta}} + \mathscr{L}_{\text{dipole}} + \mathscr{L}_{\text{Weinberg}} + \mathscr{L}_{\text{EFT}} \, . \tag{1}$$

The first term represents CP violation at mass dimension four, from the complex phases in the CKM matrix. The second arises from the gluon field strength, also at dimension four,

$$\mathscr{L}_{\bar{\theta}} = \frac{g_3^2}{32\pi^2} \bar{\theta} \, \text{Tr}(G^{\mu\nu}\tilde{G}_{\mu\nu}) \, , \tag{2}$$

where $g_3$ is the strong coupling, $G^{\mu\nu}$ is the gluon field strength, $\widetilde{G}^{\mu\nu} = \epsilon^{\mu\nu\lambda\sigma} G_{\lambda\sigma}/2$ is its dual, and $\bar{\theta}$ is the re-scaled CP-violating parameter in QCD. The bar notation indicates that corrections from the quark mass matrix are included. In principle $\bar{\theta}$ can also be included as a model parameter, for instance in testing a specific BSM model [26]. However, we take the view that the neutron EDM experimentally constrains $\bar{\theta}$ to have such a small value that this fine-tuning problem requires a proper explanation. This means there is little to be learned from including $\bar{\theta}$ as a model parameter in our global analysis.

The other three contributions in Eq.(1) are higher-dimensional and not part of a renormalizable extension of the SM-Lagrangian. Electric dipole moments of fermions $d_f^E$, and chromo-electric dipole moments of quarks $d_q^C$, appear at mass dimension five:

$$\mathscr{L}_{\text{dipole}} = -\frac{i}{2} F^{\mu\nu} \sum_{f=q,\ell} d_f^E \left( \bar{f} \sigma_{\mu\nu} \gamma_5 f \right) - \frac{i}{2} g_3 G_{\mu\nu}^a \sum_{f=q} d_q^C \left( \bar{q} \sigma^{\mu\nu} \gamma_5 T^a q \right) \, . \tag{3}$$

where the indices $q$ and $\ell$ denote quarks and leptons of all three generations. The electromagnetic field strength is $F^{\mu\nu}$. We chose the metric convention $(1, -\mathbb{1})$ with $\gamma_5 = -i\gamma_0\gamma_1\gamma_2\gamma_3$. The fermion spins are $\sigma_{\mu\nu} = i[\gamma_\mu, \gamma_\nu]/2$, and $T^a$ are the $SU(3)$ generators. The Weinberg operator is again built out of the gluon field strength and introduces the gluonic chromo-electric dipole moment $d^G$,

$$\mathscr{L}_{\text{Weinberg}} = \frac{1}{3} d^G f_{abc} G_{\mu\nu}^a \widetilde{G}^{b\nu\rho} G_\rho^{c\,\mu} \, . \tag{4}$$

Additional CP-violation occurs at mass dimension six and higher, generated at a new physics scale larger than the vacuum expectation value, $\Lambda > v$,

$$\mathscr{L}_{\text{EFT}} = \sum_i \frac{C_i^{(6)}}{\Lambda^2} \mathcal{O}_i^{(6)} + \mathcal{O}(\Lambda^{-3}) \, . \tag{5}$$

Some relevant dimension-6 operators that generate EDMs include the semileptonic and hadronic 4-fermion operators

$$\mathscr{L}_{\text{EFT}} \supset C_{\ell eqd} \left(\bar{L}^j e_R\right) \left(\bar{d}_R Q_j\right) + C_{\ell equ}^{(1)} \left(\bar{L}^j e_R\right) \epsilon_{jk} \left(\bar{Q}^k u_R\right) + C_{\ell equ}^{(3)} \left(\bar{L}^j \sigma_{\mu\nu} e_R\right) \epsilon_{jk} \left(\bar{Q}^k \sigma_{\mu\nu} u_R\right)$$
$$+ C_{quqd}^{(1)} \left(\bar{Q}^j u_R\right) \epsilon_{jk} \left(\bar{Q}^k d_R\right) + C_{quqd}^{(8)} \left(\bar{Q}^j T^a u_R\right) \epsilon_{jk} \left(\bar{Q}^k T^a d_R\right) + \text{h.c.} \tag{6}$$

The dipole moments, the Weinberg operator, and the additional 4-fermion interactions can serve as the basis for an EDM analyses in the SMEFT framework [12, 13, 17–23, 28–30]. As always, the set of higher-dimensional SMEFT operators that turn out to be most relevant depends on the high-scale BSM model that the SMEFT represents. For instance, in supersymmetric models there are no contributions from $\mathcal{O}_{quqd}^{(8)}$ and $\mathcal{O}_{\ell equ}^{(3)}$ at tree level, and the relative size of down-type and up-type quark couplings is affected by a potentially large $\tan\beta$ enhancement.

## 2.2   Hadronic-scale Lagrangian

The challenge with EDMs in view of the Lagrangian of Eq.(1) is that they are measured far below the electroweak scale, where the propagating degrees of freedom are leptons, non-relativistic nucleons $N = (p, n)^T$ with average mass $m_N$, and pions $\vec{\pi} = (\pi^+, \pi^0, \pi^-)^T$ [12, 15–17, 28, 30].

When we evolve our EFT to the experimentally relevant GeV scale, only the lepton EDMs $d_\ell^E \equiv d_\ell$ in Eq.(3) of the electroweak Lagrangian in Eq.(1) remain unchanged. While for the weak-scale Lagrangian the relation between the three charged leptons, for instance the scaling with the lepton mass, raises interesting questions, we factorize the muon and tau EDMs from the hadronic-scale Lagrangian. They can be included in the same framework, but given the systems for which experimental limits are available today, the indirect constraints on EDMs of heavy leptons are many orders of magnitude weaker than direct ones and there is little interplay among the relevant model parameters.

We split the hadronic-scale Lagrangian describing EDMs at the experimentally relevant GeV scale into

$$\mathscr{L}_{\text{had}} \supset \mathscr{L}_{N,\text{sr}} + \mathscr{L}_{\pi N} + \mathscr{L}_{eN} - \frac{i}{2} F^{\mu\nu} d_e \, \bar{e} \sigma_{\mu\nu} \gamma_5 e \, . \tag{7}$$

The dipole moments of the nucleons now read

$$\mathscr{L}_{N,\text{sr}} = -2\bar{N} \left[ d_p^{\text{sr}} \frac{1 + \tau_3}{2} + d_n^{\text{sr}} \frac{1 - \tau_3}{2} \right] S_\mu N \nu_\nu F^{\mu\nu} \, , \tag{8}$$

where $S_\mu$ and $\nu_\mu$ are the spin and velocity of the nucleon. The isoscalar and isovector contributions define $d_N^{\text{sr}}$ as the Lagrangian parameters for the short-range contributions to the proton and neutron EDMs.

Next come the interactions of pions and nucleons,

$$\mathscr{L}_{\pi N} = \bar{N} \left[ g_\pi^{(0)} \vec{\tau} \cdot \vec{\pi} + g_\pi^{(1)} \pi^0 + g_\pi^{(2)} \left(3\tau_3 \pi^0 - \vec{\tau} \cdot \vec{\pi}\right) \right] N$$
$$+ C_1 \left(\bar{N}N\right) \partial_\mu \left(\bar{N} S^\mu \bar{N}\right) + C_2 \left(\bar{N} \vec{\tau} N\right) \cdot \partial_\mu \left(\bar{N} S^\mu \bar{N} \vec{\tau}\right) + \cdots \tag{9}$$

where $\tau$ are the Pauli matrices and we neglect, for example, interactions with more than one pion. The contribution involving $g_\pi^{(2)}$ is suppressed relative to $g_\pi^{(0,1)}$ by one order in the chiral expansion [40, 41], but can be taken into account in similar fashion. In our parameterization these interactions contribute, for instance, to the nucleon EDMs through calculable pion

loops [16]. Naive dimensional analysis [42] suggests that short-range nucleon interactions enter only at NNLO [43] in the chiral expansion [44] and can be neglected [41]. In Eq.(9) this applies to all interactions in the second line. One caveat is that a consistent treatment of long-range effects may require additional short-distance counter terms [43] that appear as effective short-range nucleon-nucleon forces. These are not presently taken into account.

Finally, the higher-dimensional operators in Eq.(6) induce effective interactions that can be organized according to their tensor structure, isospin character, and their dependence on the electron and nucleon fields and spins [15, 17]:

$$
\begin{aligned}
\mathscr{L}_{eN} = &-\frac{G_F}{\sqrt{2}} \left(\bar{e}i\gamma_5 e\right) \bar{N}\left(C_S^{(0)} + C_S^{(1)}\tau_3\right)N \\
&+ \frac{8G_F}{\sqrt{2}} \, v_\nu \left(\bar{e}\sigma^{\mu\nu}e\right) \bar{N}\left(C_T^{(0)} + C_T^{(1)}\tau_3\right)S_\mu N \\
&- \frac{G_F}{\sqrt{2}} \left(\bar{e}e\right) \frac{\partial^\mu}{m_N}\left[\bar{N}\left(C_P^{(0)} + C_P^{(1)}\tau_3\right)S_\mu N\right].
\end{aligned}
\tag{10}
$$

In a heavy baryon expansion, the last line can be dropped at leading order [13]. However, we retain all three terms since (1) a pion pole enhancement of the isovector contribution somewhat offsets this hierarchy and (2) contributions of heavy quarks can also render it relevant for some new physics models. At this stage, the independent Lagrangian parameters for our global EDM analysis at the hadronic scale are:

$$
\left\{ d_e, C_S^{(0,1)}, C_T^{(0,1)}, C_P^{(0,1)}, g_\pi^{(0)}, g_\pi^{(1)}, d_{n,p}^{\mathrm{sr}} \right\}.
\tag{11}
$$

## 2.3 Matched Lagrangians for semileptonic interactions

The set of model parameters defined in Eq.(11) can be further simplified by matching the semileptonic part of the hadronic-scale Lagrangian of Eq.(10) to the corresponding weak-scale 4-fermion interactions of Eq.(6), both evaluated for external nucleons. The light quark content in the nucleons is related to the nucleon Lagrangian as

$$
\begin{aligned}
g_S^{(0)} \, \bar{\psi}_N \psi_N &= \frac{1}{2}\langle N \left|\bar{u}u + \bar{d}d\right| N \rangle \\
g_S^{(1)} \, \bar{\psi}_N \tau_3 \psi_N &= \frac{1}{2}\langle N \left|\bar{u}u - \bar{d}d\right| N \rangle \\
g_T^{(0)} \, \bar{\psi}_N \sigma_{\mu\nu} \psi_N &= \frac{1}{2}\langle N \left|\bar{u}\sigma_{\mu\nu}u + \bar{d}\sigma_{\mu\nu}d\right| N \rangle \\
g_T^{(1)} \, \bar{\psi}_N \sigma_{\mu\nu} \tau_3 \psi_N &= \frac{1}{2}\langle N \left|\bar{u}\sigma_{\mu\nu}u - \bar{d}\sigma_{\mu\nu}d\right| N \rangle \\
g_P^{(0)} \, \bar{\psi}_N \gamma_5 \psi_N &= \frac{1}{2}\langle N \left|\bar{u}\gamma_5 u + \bar{d}\gamma_5 d\right| N \rangle \\
g_P^{(1)} \, \bar{\psi}_N \gamma_5 \tau_3 \psi_N &= \frac{1}{2}\langle N \left|\bar{u}\gamma_5 u - \bar{d}\gamma_5 d\right| N \rangle,
\end{aligned}
\tag{12}
$$

which relations define the scalar, tensor, and pseudoscalar nucleon form factors $g_{S,T,P}^{(0,1)}$. Using them, the hadronic-scale Wilson coefficients in Eq.(10) can be matched to the SMEFT Wilson

coefficients in Eq.(6) as [13, 15],

$$C_S^{(0)} = -g_S^{(0)} \frac{v^2}{\Lambda^2} \operatorname{Im}\left(C_{\ell edq} - C_{\ell equ}^{(1)}\right) \qquad C_S^{(1)} = g_S^{(1)} \frac{v^2}{\Lambda^2} \operatorname{Im}\left(C_{\ell edq} + C_{\ell equ}^{(1)}\right)$$

$$C_T^{(0)} = -g_T^{(0)} \frac{v^2}{\Lambda^2} \operatorname{Im}\left(C_{\ell equ}^{(3)}\right) \qquad C_T^{(1)} = -g_T^{(1)} \frac{v^2}{\Lambda^2} \operatorname{Im}\left(C_{\ell equ}^{(3)}\right)$$

$$C_P^{(0)} = g_P^{(0)} \frac{v^2}{\Lambda^2} \operatorname{Im}\left(C_{\ell edq} + C_{\ell equ}^{(1)}\right) \qquad C_P^{(1)} = -g_P^{(1)} \frac{v^2}{\Lambda^2} \operatorname{Im}\left(C_{\ell edq} - C_{\ell equ}^{(1)}\right). \tag{13}$$

Here the six couplings $C_{S,T,P}^{(0,1)}$ are expressed in terms of only three SMEFT Wilson coefficients, implying

$$\frac{C_P^{(0)}}{g_P^{(0)}} = \frac{C_S^{(1)}}{g_S^{(1)}} \qquad \frac{C_T^{(0)}}{g_T^{(0)}} = \frac{C_T^{(1)}}{g_T^{(1)}} \qquad \frac{C_S^{(0)}}{g_S^{(0)}} = \frac{C_P^{(1)}}{g_P^{(1)}}. \tag{14}$$

Only three independent semileptonic parameters actually enter the hadronic-scale global analysis. We choose them as $C_{S,T,P}^{(0)}$ and combine them using the known ratios of hadronic matrix elements to construct the full Lagrangian. In addition to the light quarks described by Eq.(13), the relations in Eq.(14) also have to include the contributions of heavy quarks. These contributions are contained in the nucleon form factors, renormalized at an appropriate mass scale and accounting for the corresponding anomaly relations.

Note that in this way our global analysis preserves the full dependence on $C_{S,T,P}^{(0,1)}$, although only $C_{S,T,P}^{(0)}$ appear as model parameters. This remains the case regardless of isospin-violating effects, and also when including experimental systems for which the coefficients of $C_S^{(1)}$ differ significantly, as discussed below.

The implementation of Eq.(13) for $C_S^{(0,1)}$ is particularly straightforward, because for each experimental system the effective parameter that combines the isoscalar and isovector terms is independent of the nuclear spin,

$$C_S = C_S^{(0)} + \frac{Z-N}{Z+N} C_S^{(1)}$$

$$= C_S^{(0)} + \frac{Z-N}{Z+N} \frac{g_S^{(1)}}{g_P^{(0)}} C_P^{(0)} \qquad \text{with} \qquad \frac{g_S^{(1)}}{g_P^{(0)}} \approx 0.1. \tag{15}$$

In the second step we replace $C_S^{(1)}$ with $C_P^{(0)}$, reflecting Eq.(13). Since $g_S^{(1)}$ is already suppressed relative to $g_S^{(0)}$ by the small isospin violation of the nucleon matrix element, one could argue that $C_S^{(1)} \ll C_S^{(0)}$. Moreover, in the heavy nuclei of all atomic and molecular systems for which EDMs have been measured so far, the isoscalar and isovector contributions occur in approximately the same ratio, $(Z-N)/(Z+N) \approx -0.2$, so the effective parameter $C_S$ is approximately system-independent. As noted above, we do not rely on these assumptions.

Next, we relate the pseudoscalar and tensor semileptonic interactions in a similar fashion. We start with the linear combinations for nucleons, $C_{P,T}^{(n,p)} = C_{P,T}^{(0)} \mp C_{P,T}^{(1)}$, taking the particle physics convention $\tau_3 |n\rangle = -|n\rangle$). From those, the coefficients for a given nucleus can be constructed according to the sum over spins of the constituent nucleons, where $\langle \sigma_{p,n} \rangle$ represents the average neutron or proton spin, evaluated for the measured nuclear state [45, 46].

$$C_{P,T} = \frac{C_{P,T}^{(n)} \langle \sigma_n \rangle + C_{P,T}^{(p)} \langle \sigma_p \rangle}{\langle \sigma_n \rangle + \langle \sigma_p \rangle}. \tag{16}$$

For $C_T$ we can see from Eq.(13) that the isoscalar and isovector couplings differ only through the corresponding nucleon form factors. These are calculated with small theoretical uncertainties in lattice QCD [47], allowing us to write

$$C_T = \left(1 - \frac{g_T^{(1)}}{g_T^{(0)}} \frac{\langle\sigma_n\rangle - \langle\sigma_p\rangle}{\langle\sigma_n\rangle + \langle\sigma_p\rangle}\right) C_T^{(0)} \qquad \text{with} \qquad \frac{g_T^{(1)}}{g_T^{(0)}} \approx 1.7 \,. \tag{17}$$

Similarly, for $C_P$ it can be shown that

$$C_P = C_P^{(0)} - \frac{g_P^{(1)}}{g_S^{(0)}} \frac{\langle\sigma_n\rangle - \langle\sigma_p\rangle}{\langle\sigma_n\rangle + \langle\sigma_p\rangle} C_S^{(0)} \qquad \text{with} \qquad \frac{g_P^{(1)}}{g_S^{(0)}} \approx 20.2 \,. \tag{18}$$

For the derivation of $g_P^{(1)}$ in this ratio we follow Ref. [13]. We consider only the first generation as relevant light quarks, such that $g_P^{(1)}$ is dominated by the $\pi$-pole contribution

$$g_P^{(1)} = \frac{g_A \bar{m}_N}{\bar{m}} \frac{m_\pi^2}{m_\pi^2 - q^2} + \text{heavy quarks} \qquad \bar{m} = \frac{m_u + m_d}{2} \,. \tag{19}$$

Here $\bar{m}_N \approx 940$ MeV is the average nucleon mass, $\bar{m}$ the average light quark mass, and $g_A$ is the (isovector) axial vector coupling [48]. The coupling $g_P^{(0)}$, appearing in Eq.(15), involves the isoscalar axial coupling $g_A^{(0)}$, obtained from the sum rather than the difference of the light quark axial charges. We extend this by including a light $s$-quark, where the $\pi$-pole dominance is replaced by an octet $\eta$-pole with an appropriately modified average light quark mass $m^*$

$$g_P^{(0)} = \frac{g_A^{(0)} \bar{m}_N}{m^*} \frac{m_\eta^2}{m_\eta^2 - q^2} + \text{heavy quarks} \qquad m^* = \frac{m_u + m_d + 2m_s}{3} \,. \tag{20}$$

Note that heavy quark contributions enter differently in $g_P^{(0)}$ and $g_P^{(1)}$, and can be derived using the $U(1)_A$ axial anomaly together with the divergence of the anomaly-free axial current $J_{\mu 5}^q = \bar{q}\gamma_\mu\gamma_5 q$ for all quarks $q$ [49–52]. These represent a relatively minor contribution to $g_P^{(1)}$, but due to suppression of the $\eta$-pole relative to the $\pi$-pole by the factor $m^*/\bar{m}$, contribute to $g_P^{(0)}$ at approximately the same level as the light quarks.

Our final, simplifying assumption is not strictly needed, but is effective in reducing the number of model parameters by one,

$$d_p^{\text{sr}} \approx -d_n^{\text{sr}} \,. \tag{21}$$

The nuclei of diamagnetic systems, which apart from the neutron itself provide the strongest constraints on nucleon EDMs, are typically dominated either by a valence proton or a valence neutron. As for the anomalous magnetic moment, the short-range nucleon EDMs are assumed to be dominated by the isovector contribution. This assumption can be relaxed for the purposes of a rigorous global analysis, provided improved theory uncertainties concerning the contributions of all nucleons to the overall EDM of each measured system. Of the present experimental limits, $d_p^{\text{sr}}$ is constrained essentially only by the TlF measurement.

With these simplifications, the set of low-energy parameters given in Eq.(11) reduces to

$$c_j \in \left\{ d_e, C_S^{(0)}, C_T^{(0)}, C_P^{(0)}, g_\pi^{(0)}, g_\pi^{(1)}, d_n^{\text{sr}} \right\} \,. \tag{22}$$

These Lagrangian parameters define the model parameters for our global EDM analysis.

## 3   EDM Measurements

In terms of the Lagrangian parameters in Eq.(22) we can predict the measured EDMs $d_i$ as linear combinations with system-specific coefficients $\alpha_{i,c_j}$,

$$d_i = \sum_{c_j} \alpha_{i,c_j} c_j \,. \tag{23}$$

The measurements we analyze are listed in Tab. 1 and discussed below. Unless otherwise indicated, the isotopes and charge states that we discuss are those given in Tab. 1. The isotopes that are relevant for these molecular systems are $^{180}$Hf, $^{232}$Th, $^{174}$Yb, $^{205}$Tl, $^{16}$O, and $^{19}$F. We neglect constraints from the comparatively weaker experimental bounds from $^{85}$Rb [53, 54], Xe$^m$ [55], PbO [56], Eu$_{0.5}$Ba$_{0.5}$TiO$_3$ [57], and the $\Lambda$ hyperon [58]. The experimental bounds for the $\mu$ and $\tau$ leptons constrain the corresponding Lagrangian parameters and factorize from the hadronic-scale Lagrangian. We do not include them in our first global analysis.

As discussed in the last section, the $\alpha$-values for all $C_{S,P,T}^{(0,1)}$ can be extracted from the corresponding values for the basis given in Eq.(22). For this we follow Eqs. (15)-(18) and find for example

$$
\begin{aligned}
\alpha_{C_S^{(0)}} &= \alpha_{C_S} - \alpha_{C_P} \frac{g_P^{(1)}}{g_S^{(0)}} \frac{\langle \sigma_n \rangle - \langle \sigma_p \rangle}{\langle \sigma_n \rangle + \langle \sigma_p \rangle} \\
\alpha_{C_P^{(0)}} &= \alpha_{C_P} + \alpha_{C_S} \frac{g_S^{(1)}}{g_P^{(0)}} \frac{Z - N}{Z + N} \\
\alpha_{C_T^{(0)}} &= \left( 1 - \frac{g_T^{(1)}}{g_T^{(0)}} \frac{\langle \sigma_n \rangle - \langle \sigma_p \rangle}{\langle \sigma_n \rangle + \langle \sigma_p \rangle} \right) \alpha_{C_T} \,,
\end{aligned} \tag{24}
$$

with the $\alpha_{C_{S,P,T}}$ given in Tab. 2.

### 3.1   Nucleons

We start with the simplest hadronic systems for which we can measure EDMs, the nucleons. Their EDMs can be written directly as a Lagrangian term

$$-\frac{i}{2} F^{\mu\nu} d_N \left( \bar{N} \sigma_{\mu\nu} \gamma_5 N \right) \,, \tag{25}$$

but can also be described by chiral perturbation theory, based on the hadronic-scale Lagrangian. In that case we consider the nucleon EDMs as observables that receive contributions from: short-range contributions $d_N^{\text{sr}}$, NNLO pion-loop contributions, and potential direct contributions to Eq.(25) within and beyond the SM [73],

$$d_n = d_n^{\text{sr}} - \frac{e g_A}{8\pi^2 F_\pi} \left[ g_\pi^{(0)} \left( \ln \frac{m_\pi^2}{m_N^2} - \frac{\pi m_\pi}{2 m_N} \right) - \frac{g_\pi^{(1)}}{4} (\kappa_0 - \kappa_1) \frac{m_\pi^2}{m_N^2} \ln \frac{m_\pi^2}{m_N^2} \right] \tag{26}$$

$$d_p = d_p^{\text{sr}} + \frac{e g_A}{8\pi^2 F_\pi} \left[ g_\pi^{(0)} \left( \ln \frac{m_\pi^2}{m_N^2} - \frac{2\pi m_\pi}{m_N} \right) - \frac{g_\pi^{(1)}}{4} \left( \frac{2\pi m_\pi}{m_N} + \left( \frac{5}{2} + \kappa_0 + \kappa_1 \right) \frac{m_\pi^2}{m_N^2} \ln \frac{m_\pi^2}{m_N^2} \right) \right] \,,$$

where $F_\pi = 92$ MeV is the pion decay constant [74], $m_\pi = 139$ MeV, $m_N = 940$ MeV, $g_A \approx 1.27$ is the nucleon isovector axial charge, and the isoscalar and isovector nucleon anomalous magnetic moments are $\kappa_0 = -0.12$ and $\kappa_1 = 3.7$. We set the renormalization scale to the nucleon

| System $i$ | Measured $d_i$ [$e$ cm] | Upper limit on $|d_i|$ [$e$ cm] | Reference |
|---|---|---|---|
| $n$ | $(0.0 \pm 1.1_{\text{stat}} \pm 0.2_{\text{syst}}) \cdot 10^{-26}$ | $2.2 \cdot 10^{-26}$ | [59] |
| $^{205}$Tl | $(-4.0 \pm 4.3) \cdot 10^{-25}$ | $1.1 \cdot 10^{-24}$ | [60] |
| $^{133}$Cs | $(-1.8 \pm 6.7_{\text{stat}} \pm 1.8_{\text{syst}}) \cdot 10^{-24}$ | $1.4 \cdot 10^{-23}$ | [61] |
| HfF$^+$ | $(-1.3 \pm 2.0_{\text{stat}} \pm 0.6_{\text{syst}}) \cdot 10^{-30}$ | $4.8 \cdot 10^{-30}$ | [62] |
| ThO | $(4.3 \pm 3.1_{\text{stat}} \pm 2.6_{\text{syst}}) \cdot 10^{-30}$ | $1.1 \cdot 10^{-29}$ | [63] |
| YbF | $(-2.4 \pm 5.7_{\text{stat}} \pm 1.5_{\text{syst}}) \cdot 10^{-28}$ | $1.2 \cdot 10^{-27}$ | [64] |
| $^{199}$Hg | $(2.20 \pm 2.75_{\text{stat}} \pm 1.48_{\text{syst}}) \cdot 10^{-30}$ | $7.4 \cdot 10^{-30}$ | [65, 66] |
| $^{129}$Xe | $(-1.76 \pm 1.82) \cdot 10^{-28}$ | $4.8 \cdot 10^{-28}$ | [67, 68] |
| $^{171}$Yb | $(-6.8 \pm 5.1_{\text{stat}} \pm 1.2_{\text{syst}}) \cdot 10^{-27}$ | $1.5 \cdot 10^{-26}$ | [69] |
| $^{225}$Ra | $(4 \pm 6_{\text{stat}} \pm 0.2_{\text{syst}}) \cdot 10^{-24}$ | $1.4 \cdot 10^{-23}$ | [70] |
| TlF | $(-1.7 \pm 2.9) \cdot 10^{-23}$ | $6.5 \cdot 10^{-23}$ | [71] |

| | Measured $\omega_i$ [mrad/$s$] | Rescaling factor $x_i$ for $d_i$ | Reference |
|---|---|---|---|
| HfF$^+$ | $(-0.0459 \pm 0.0716_{\text{stat}} \pm 0.0217_{\text{syst}})^\dagger$ | 0.999 | [62] |
| ThO | $(-0.510 \pm 0.373_{\text{stat}} \pm 0.310_{\text{syst}})$ | 0.982 | [63] |
| YbF | $(5.30 \pm 12.60_{\text{stat}} \pm 3.30_{\text{syst}})$ | 1.12 | [64] |

Table 1: Measured EDM values and 95%CL ranges. For $^{129}$Xe we combine two independent results with similar precision, using inverse-variance weighting. For the paramagnetic molecules, we also provide the measured angular frequencies and the rescaling factor which allows us to use $x_i d_i$ for each experimentally reported $d_i$. For the definition of $x_i$, see text. $^\dagger$The frequency for HfF$^+$ is scaled by a factor of 2 relative to Ref. [62], to consistently use Eq.(29) for all systems.

mass, and the split between proton and neutron mass leads to a higher-order effect that is presently negligible in relation to other uncertainties. In terms of weak-scale parameters, finite values of $g_\pi^{(0,1)}$ can be related to a CKM phase or $\bar\theta$, but also to hadronic 4-fermion operators. The input from the presently most sensitive neutron EDM measurement to our global analysis is given in Tab. 1.

## 3.2 Paramagnetic systems

Paramagnetic atoms and molecules are primarily sensitive to the electron EDM and the scalar electron-nucleon couplings $C_S^{(0,1)}$ of Eq.(10). It is common to distinguish the case of paramagnetic atoms with the atomic EDMs

$$d_i = \alpha_{i,d_e} d_e + \alpha_{i,C_S} C_S + \sum_{c_j} \alpha_{i,c_j} c_j \qquad \text{for} \qquad i \in \{\text{Tl}, \text{Cs}\} \tag{27}$$

from that of paramagnetic molecules, although in both cases the terms involving the Lagrangian parameters $d_e$ and $C_S$ dominate. The experimentally relevant quantity is a phase difference accumulated over some measurement time, interpreted as a frequency. Measurements with paramagnetic molecules are often reported as a parity- and time-reversal-violating frequency shift $\omega_i$,

$$\omega_i = \eta_{i,d_e}^{(m)} d_e + k_{i,C_S}^{(m)} C_S + \sum_{c_j} \alpha_{i,c_j}^{(m)} c_j \qquad \text{for} \qquad i \in \{\text{HfF}^+, \text{ThO}, \text{YbF}\} . \tag{28}$$

The superscript $(m)$ indicates the molecular systems. We also give these measured angular frequencies in Tab. 1. This emphasizes that $\omega_i$ does not scale in a simple way with the experimentally applied electric field; it rather depends on the molecular structure via a so-called

| System $i$ | $\langle\sigma_n\rangle$ | $\langle\sigma_p\rangle$ | $\langle\sigma_z\rangle^{(0)}$ | $\alpha_{i,C_S}$ [$e$ cm] | $\alpha_{i,C_P}$ [$e$ cm] | $\alpha_{i,C_T}$ [$e$ cm] |
|---|---|---|---|---|---|---|
| Tl | 0.274 | 0.726 | 1 | $-6.77\cdot10^{-18}$ | $1.5\cdot10^{-23}$ | $5\cdot10^{-21}$ |
| Cs | $-0.206$ | $-0.572$ | $-0.778$ | $7.8\cdot10^{-19}$ | $2.2\cdot10^{-23}$ | $9.2\cdot10^{-21}$ |
| $^{199}$Hg | $-0.302$ | $-0.032$ | $-0.334$ | $-2.8\cdot10^{-22}$ | $6\cdot10^{-23}$ | $1.7\cdot10^{-20}$ |
| $^{129}$Xe | 0.73 | 0.27 | 1 | $-6.28\cdot10^{-23}$ | $1.6\cdot10^{-23}$ | $5.7\cdot10^{-21}$ |
| $^{171}$Yb | $-0.3$ | $-0.034$ | $-0.334$ | $-2.68\cdot10^{-22}$ | $4.01\cdot10^{-23}$ | $1.24\cdot10^{-20}$ |
| $^{225}$Ra | 0.72 | 0.28 | 1 | $2.9\cdot10^{-21}$ | $-6.4\cdot10^{-22}$ | $-1.8\cdot10^{-19}$ |
| TlF | 0.274 | 0.726 | 1 | $2.9\cdot10^{-18}$ | $3\cdot10^{-19}$ | $2.7\cdot10^{-16}$ |

Table 2: Effective parameters used as input to our global analysis, as summarized in Tab. 3. Within a shell model of the nucleus, the quantity $\langle\sigma_z\rangle^{(0)} = \langle\sigma_n\rangle + \langle\sigma_p\rangle$ is the isoscalar sum of neutron and proton spin projections. (Note that the shell model is not expected to be reliable for the deformed nuclei $^{171}$Yb and $^{225}$Ra.) The spin fractions contributing to semileptonic coefficients in TlF only take into account the $^{205}$Tl nucleus, though see Ref. [72] for some consideration of contributions from the $^{19}$F nucleus.

effective electric field $E_{\text{eff}}$ that saturates when the molecule is polarized. Note that $\alpha_{i,d_e}$ is dimensionless, while $\eta^{(m)}_{i,d_e}$ is not; similarly $\alpha_{i,C_S}$ has the units of an EDM, while $k^{(m)}_{i,C_S}$ has units of angular frequency.

The remaining contributions to paramagnetic molecule EDMs have weak dependences on other low-energy constants, and we note that in particular the EDMs of the $\mu$ and $\tau$ leptons have been indirectly constrained via the experimental limits from ThO and Hg [91]. However, we are not aware of any established values for the coefficients of semileptonic or hadronic parameters. We thus take $\alpha^{(m)}_{i,c_j} = 0$ in Eq.(28), while for the paramagnetic atoms we obtain values for semileptonic coefficients either from the literature or by scaling arguments, as reported in Tabs. 2 and 3.

Now starting with the truncated version of Eq.(28), we adopt the convention that

$$
\begin{aligned}
\omega_i &= \eta^{(m)}_{i,d_e}d_e + k^{(m)}_{i,C_S}C_S \\
&= -\frac{E_{\text{eff},i}}{\hbar}d_e + \frac{W_i}{\hbar}C_S \, ,
\end{aligned}
\tag{29}
$$

where all signs, $g$-factors, spin magnitudes, etc. are absorbed into the coefficients on the right-hand side. It is also common to refer to $\mathcal{E}_{\text{eff}}$ as the effective electric field, where

$$
E_{\text{eff}} = \mathcal{E}_{\text{eff}}\,\text{sgn}(\vec{J}\cdot\hat{n})\,\langle\hat{n}\cdot\hat{z}\rangle \, ,
\tag{30}
$$

and $\vec{J}$ is the total electronic angular momentum, $\hat{n}$ is the direction of the internuclear axis, and $\hat{z}$ is the direction of the externally applied electric field. Here $\Omega = \vec{J}\cdot\hat{n}$ is a quantum number used in molecular term symbols, and $\langle\hat{n}\cdot\hat{z}\rangle$ indicates the degree to which the molecule is electrically polarized by the applied electric field. Some of the conventions used for the orientation of the internuclear axis, the internal electric field, etc. are summarized in Appendix A.3 of Ref. [92].

The truncated expression for the frequency shift thus provides us with physical expressions for the two constants,

$$
\eta^{(m)}_{i,d_e} = -\frac{E_{\text{eff},i}}{\hbar} \qquad \text{and} \qquad k^{(m)}_{i,C_S} = \frac{W_i}{\hbar} \, .
\tag{31}
$$

| System $i$ | $\alpha_{i,d_e}$ | $\alpha_{i,C_S}^{(0)}$ [$e$ cm] | $\alpha_{i,C_P}^{(0)}$ [$e$ cm] | $\alpha_{i,C_T}^{(0)}$ [$e$ cm] | $\alpha_{i,g_\pi}^{(0)}$ [$e$ cm] | $\alpha_{i,g_\pi}^{(1)}$ [$e$ cm] | $\alpha_{i,d_n}^{\mathrm{str}}$ | $\alpha_{i,d_p}^{\mathrm{str}}$ |
|---|---|---|---|---|---|---|---|---|
| $n$ | — | — | — | — | $1.38^{\pm0.02} \cdot 10^{-14}$ | $2.73^{\pm0.02} \cdot 10^{-16}$ | $1$ | $-1$ |
| $^{205}$Tl | $-558^{\pm28}$ [75] | $-6.77^{\pm0.34} \cdot 10^{-18}$ | $1.5^{+2}_{-0.7} \cdot 10^{-19}$ | $8.8^{\pm0.9} \cdot 10^{-21}$ | n/a | n/a | n/a | n/a |
| $^{133}$Cs | $123^{\pm4}$ | $7.80^{+0.2}_{-0.8} \cdot 10^{-19}$ | $-1.4^{+0.7}_{-2} \cdot 10^{-20}$ | $1.7^{\pm0.2} \cdot 10^{-20}$ | — | — | — | — |
| $^{199}$Hg | $-0.012^{+0.0094}_{-0.002}$ [33,76] | $-1.26^{+0.7}_{-1.2} \cdot 10^{-21}$ | $6.6^{+1.2}_{-0.3} \cdot 10^{-23}$ | $-6.4^{+3}_{-2.62} \cdot 10^{-21}$ | $-1.18^{+0.19}_{-2.62} \cdot 10^{-17}$ | $1.6^{+0}_{-6.5} \cdot 10^{-17}$ | $-1.56^{\pm0.39} \cdot 10^{-4}$ | $-1.56^{\pm0.39} \cdot 10^{-5}$ |
| $^{129}$Xe | $-8^{+0}_{-8} \cdot 10^{-4}$ [33,77] | $-2.1^{-1.2} \cdot 10^{-22}$ | $1.7^{+0.5}_{-0.4} \cdot 10^{-23}$ | $1.24^{+0.78}_{-0.61} \cdot 10^{-21}$ | $-0.4^{+1.2}_{-23} \cdot 10^{-19}$ | $-2.2^{+1.1}_{-0} \cdot 10^{-19}$ | $1.7^{+0.7}_{-0} \cdot 10^{-5}$ | $3.51^{\pm0.88} \cdot 10^{-6}$ |
| $^{171}$Yb | $(-0.012^{+0.01145}_{-0.002})$ [78] | $-9.1^{-5} \cdot 10^{-22}$ | $4.5^{+1.8}_{-2.9} \cdot 10^{-23}$ | $-4.4^{+2.2}_{-2.0} \cdot 10^{-21}$ | $-9.5^{+23}_{-2.4} \cdot 10^{-18}$ | $1.3^{+0.33}_{-0} \cdot 10^{-17}$ | $-1.13^{\pm0.28} \cdot 10^{-4}$ | $-1.13^{\pm0.28} \cdot 10^{-5}$ |
| $^{225}$Ra | $-0.054^{\pm0.002}$ [33] | $8.6^{+9.5}_{-4.5} \cdot 10^{-21}$ | $-7.0^{+1.7}_{-1.1} \cdot 10^{-22}$ | $-4.5^{+2.0}_{-2.5} \cdot 10^{-20}$ | $1.7^{+5.2}_{-0.8} \cdot 10^{-15}$ | $-6.9^{+3.1}_{-21} \cdot 10^{-15}$ | $-5.36^{\pm1.34} \cdot 10^{-4}$ | $-1.11^{\pm0.28} \cdot 10^{-4}$ |
| TlF | $81^{\pm20}$ [51,71] | $5.6^{-4.5}_{-2.5} \cdot 10^{-18}$ | $2.4^{+1.0}_{-1.9} \cdot 10^{-19}$ | $4.8^{+1.2}_{-1.1} \cdot 10^{-16}$ | $1.9^{+0.1}_{-1.4} \cdot 10^{-14}$ | $-1.6^{+0.4}_{-0} \cdot 10^{-13}$ | $-9.47^{\pm2.37} \cdot 10^{-2}$ | $-4.59^{\pm1.15} \cdot 10^{-1}$ |
| HfF$^+$ | $1$ | $9.17^{\pm0.06} \cdot 10^{-21}$ | — | — | — | — | — | — |
| ThO | $1$ | $1.51^{+0}_{-0.2} \cdot 10^{-20}$ | — | — | — | — | — | — |
| YbF | $1$ | $8.99^{\pm0.70} \cdot 10^{-21}$ | — | — | — | — | — | — |

| System $i$ | $\eta_{i,d_e}^{(m)} \left[ \dfrac{\mathrm{mrad}}{\mathrm{s}\,e\,\mathrm{cm}} \right]$ | $\kappa_{i,C_S}^{(m)} \left[ \dfrac{\mathrm{mrad}}{\mathrm{s}} \right]$ | $\alpha_{i,C_P}$ | $\alpha_{i,C_T}$ | $\alpha_{i,g_\pi}^{(0)}$ | $\alpha_{i,g_\pi}^{(1)}$ | $\alpha_{i,d_n}^{\mathrm{str}}$ | $\alpha_{i,d_p}^{\mathrm{str}}$ |
|---|---|---|---|---|---|---|---|---|
| HfF$^+$ | $3.49^{\pm0.14} \cdot 10^{28}$ [76,79–82] | $3.2^{+0.1}_{-0.2} \cdot 10^8$ [76,79,80] | — | — | — | — | — | — |
| ThO | $-1.21^{+0.05}_{-0.39} \cdot 10^{29}$ [76,83–85]† | $-1.82^{+0.42}_{-0.27} \cdot 10^9$ [76,83,85–87]† | — | — | — | — | — | — |
| YbF | $-1.96^{\pm0.15} \cdot 10^{28}$ [76,86–89] | $-1.76^{\pm0.2} \cdot 10^8$ [76,86–88] | — | — | — | — | — | — |

Table 3: Central values for the $\alpha$-parameters defined in Eq.(23). Here n/a means that we do not know of a reliable prediction, and — means that we neglect the dependence in our global analysis. †There appears to be an overall sign issue in the coefficients reported for ThO in Table 4 of Ref. [76]. For the coefficient of $d_e$ in $^{171}$Yb, we take the value from the structurally similar system $^{199}$Hg, since no complete dedicated calculation is presently available (the result of [78] is taken into account via the upper error bound). The impact of this choice on the global analysis is negligible. The values for Tl and Cs of $\alpha_{i,C_T}^{(0)}$ are estimated by simple analytical calculations [90], and the uncertainties quoted here are estimated as approximately twice those arising from the relevant hadronic matrix elements.

The point is that this frequency measurement cannot be directly translated into a permanent molecular dipole moment, since the molecules are substantially or entirely electrically polarized during the measurement. The linear energy shift due to the lab-frame electric field saturates as $\langle \hat{n} \cdot \hat{z} \rangle \to 1$, and the limiting dependence on the external fields is that of an induced dipole moment. To express the frequency difference $\omega_i$ in terms of a physically meaningful permanent electric dipole moment, we define the molecule's $d_i$ in relation to the effective electric field,

$$-E_{\text{eff},i} d_i \equiv -E_{\text{eff},i} d_e + W_i C_S . \tag{32}$$

We can use this form to define, in analogy to Eq.(27),

$$d_i = d_e + \alpha_{i,C_S} C_S \qquad \Leftrightarrow \qquad \alpha_{i,d_e} = 1 \quad \text{and} \quad \alpha_{i,C_S} = -\frac{W_i}{E_{\text{eff},i}} = \frac{k_{i,C_S}^{(m)}}{\eta_{i,d_e}^{(m)}} . \tag{33}$$

For our analysis we re-cast the limits from paramagnetic molecules in units of $e$cm, such that the coefficients for all systems can be expressed in the same units, with the side effect that $\alpha_{i,d_e} = 1$ for all paramagnetic molecules. This approach also serves two additional purposes: (i) Many different conventions are in use for these coefficients, sometimes using the same symbols for different quantities. Since measured quantities are related in publications to the electron EDM, our choice makes comparisons across different works relatively straightforward. Of course, all quantities must be ultimately connected to an experimentally measured phase; (ii) As first pointed out in Refs. [86,87], while there is considerable variation in the literature values for $\eta_{i,d_e}^{(m)}$ and $k_{i,C_S}^{(m)}$ in a given system, there is much less variation in their ratio. Dividing Eq.(28) by $\eta_{i,d_e}^{(m)}$ should pass this uncertainty on to all semileptonic and hadronic coefficients.

Note that $\eta_{i,d_e}^{(m)}$ is used both to obtain sole-source limits on $d_e$ from the experimentally measured $\omega_i$, and to convert the experimentally measured frequencies into EDM units for Tab. 1. The experimental EDM limits for paramagnetic molecules in Tab. 1 are rescaled by the indicated factor $x_i$, which differs from unity when updated values for $\eta_{i,d_e}^{(m)}$ differ from the cited publication. This is typically the case when improved molecular structure calculations have become available since the experimental limit was published.

Recommended values for many of the $\alpha_{i,c_j}$ are given in Tables III-V of Ref. [15] and in Table 4 of Ref. [76], including in many cases the ranges corresponding to theory uncertainties (or at least different reported values). We give our choices for $\alpha_{i,c_j}$ in Tab. 3.

## 3.3 Diamagnetic systems

In contrast to the paramagnetic systems, where sub-leading contributions are largely neglected due to lack of theory inputs, the small contributions of $d_e$ and $C_S^{(0)}$ are taken into account for all diamagnetic systems. These are typically smaller contributions to the observable EDM in cases where all electron spins are paired, with the main contributions coming rather from nucleon EDMs, nuclear forces mediated by pion exchange with strengths $g_\pi^{(0,1,2)}$, and the nuclear-spin-dependent semileptonic interactions $C_T^{(0)}$ (and possibly $C_P^{(0)}$). The experimental precision, especially of Hg, is nevertheless high enough to contribute meaningful constraining power for $d_e$ and $C_S^{(0)}$.

Unfortunately it appears that arguments [76] to the effect that diamagnetic systems add significant complementary constraining power in the $d_e - C_S$ subspace due to a different sign of the ratio $\alpha_{i,d_e}/\alpha_{i,C_S}$ in comparison to paramagnetic systems, do not in fact hold for Hg or

most other systems in Tab. 1. On the other hand, recent calculations [33] indicate that $\alpha_{i,d_e}$ and $\alpha_{i,C_S}$ in some highly correlated systems may indeed have different sign, thus reviving this possibility. At present, and as can be seen from Tab. 3, Ra is the only measured system where this appears to be the case. It may thus represent a special case among diamagnetic systems, in that reducing the error of experimental and theory inputs could have significant impact on the $d_e - C_S^{(0)}$ subspace that is complementary to the dominating paramagnetic constraints.

The contributions of nuclear forces and nucleon EDMs are frequently interpreted via the Schiff moment of a given nucleus [15], which is more easily related to nuclear structure parameters [90]. In terms of our model coefficients, the Schiff moment $S_i$ of system $i$ and a system-specific coefficient $k_{i,S}$ can be expressed related to the corresponding EDM via

$$
\begin{aligned}
k_{i,S} S_i &= \sum_{c_j \in \{d_{n,p}^{\mathrm{sr}}, g_\pi^{(0,1,2)}\}} \alpha_{i,c_j} c_j \\
&= k_{i,S} \left[ s_{i,n} d_n + s_{i,p} d_p + \frac{m_N g_A}{F_\pi} \left( a_{i,0} g_\pi^{(0)} + a_{i,1} g_\pi^{(1)} + a_{i,2} g_\pi^{(2)} \right) \right] \\
&\approx k_{i,S} \left[ s_{i,n} d_n^{\mathrm{sr}} + s_{i,p} d_p^{\mathrm{sr}} + \frac{m_N g_A}{F_\pi} \left( \tilde{a}_{i,0} g_\pi^{(0)} + \tilde{a}_{i,1} g_\pi^{(1)} \right) \right],
\end{aligned}
\tag{34}
$$

where the coefficients $s_{i,N}$ ($N = n, p$) indicate the contributions from EDMs of unpaired nucleons and the coefficients $a_{i,m}$ ($m = 0, 1, 2$) parameterize the strength of CP-violating pion exchange, organized by isospin, for the nucleus of system $i$. Between the second and third lines we drop the $g_\pi^{(2)}$ term as discussed above, and use Eq.(26) to rewrite the nucleon EDMs themselves and absorb their contributions from $g_\pi^{(0)}$ and $g_\pi^{(1)}$ into the implicitly re-defined coefficients $\tilde{a}_{i,m}$.

The coefficient $k_{i,S}$ is calculated for many systems of interest and can, in principle, be large in heavy and especially deformed nuclei such as $^{225}$Ra. This corresponds to a nuclear-structure induced enhancement of the observable EDM, which applies to both the nucleon EDMs and the pion-exchange forces. This implies that among the potentially leading contributions from the six hadronic-scale model parameters $C_{P,T}^{(0)}$, $d_{n,p}^{\mathrm{sr}}$, and $g_\pi^{(0,1)}$, very different weights can be expected according to the electronic and nuclear structure of the various diamagnetic systems. Inspecting Tab. 3 reveals that to a limited extent, this is indeed already the case. However, this complementarity is not yet fully exploited.

Despite the pion pole enhancement, $C_P^{(0)}$ appears suppressed relative to $C_T^{(0)}$ in all measured systems. The coefficients of $g_\pi^{(0,1)}$ are typically of comparable size for a given system, although possibly different in sign. Unfortunately the coefficients of $d_{n,p}$ are not well known for most nuclei, although in principle these can be calculated. For the pion-exchange forces, even in nuclei that have been the object of many studies, the theory uncertainties are large. This is especially true for soft nuclei such as our Hg and Xe, in which non-static deformations can present special challenges.

For nuclei with spin $I > 1/2$, a CP-violating nuclear magnetic quadrupole moment can in principle exist in analogy to a Schiff-moment-induced EDM and be analyzed within a common global analysis. At present among the experimental systems which have already been measured, Cs is the one where this effect would be expected to be most relevant.

As part of a global analysis, a large number of experimental measurements from complementary diamagnetic systems can disentangle contributions from model parameters that are relevant at some level for all of them. In this sense the role of $d_e$ and $C_S^{(0)}$ takes on a new importance, not only to further constrain these parameters themselves, but also as an additional contribution to the EDM that brings along uncertainties which dilute the constraining

power for other model parameters. Diamagnetic molecular systems such as TlF, or molecular systems containing Schiff-enhanced nuclides, introduce complementary constraining power to our global analysis.

We finally note that diamagnetic systems unite theory inputs from different communities. Different conventions for assigning a negative isospin projection in the nucleon doublet have to be carefully noted when combining calculated coefficients from different sources, especially when we include semileptonic interactions together with nuclear forces. Reference [93] makes an effort to disambiguate a part of this issue for $\alpha_{g_\pi^{(0,1,2)},^{129}Xe}$.

It is only possible to establish meaningful constraints in a context where the notation and conventions are clear, and where the uncertainties associated with theory inputs have been clearly assessed.

### 3.4 Single parameter ranges

In this section we provide toy ranges of individual model parameters from the set

$$\left\{ d_e, C_S^{(0)}, C_T^{(0)}, C_P^{(0)}, g_\pi^{(0)}, g_\pi^{(1)}, d_n^{\mathrm{sr}} \right\}, \tag{35}$$

using the set of EDM measurements one by one. To extract these single parameter ranges, we utilize Eq.(27) and include all EDM measurements listed in Tab. 1 and the model dependence from Tab. 3. For the likelihood we assume a Gaussian form, which means that all limits are quoted as symmetric one-sigma error bars around the central, best-fit value. While these single-parameter limits allow us to compare the reach of different measurements for CP violation as a whole, they do not give the allowed ranges of the individual model parameters. The reason is that contributions from different model parameters to the same measurement can cancel. Because an EFT builds on the assumption that many higher-dimensional operators are induced by a given physics model, the single-parameter limits are overly optimistic.

Using the single-parameter constraints in Tab. 4 we can compare the impact of different EDM measurements on a given model parameter, with the caveat that multi-dimensional correlations might change that picture. Starting with the electron EDM $d_e$, the paramagnetic molecules HfF$^+$ and ThO provide the strongest constraints, while the paramagnetic YbF molecule has a similar constraining power as the diamagnetic Hg. The same two paramagnetic molecules lead the constraining power for the scalar electron-nucleon coupling $C_S^{(0)}$, again followed by Hg, YbF, and Tl.

The pseudoscalar and tensor electron-nucleon couplings $C_{P,T}^{(0)}$ can be probed by the paramagnetic atoms and the diamagnetic systems. This is done at present by far most efficiently with Hg and then, with much reduced constraining power, by Tl, Xe, and TlF. It is difficult to estimate the impact of these measurements on the combination of $C_P^{(0)}$ and $C_T^{(0)}$, and we will see in the next section how this more complex dependence affects the full global analysis.

The three hadronic parameters, $g_\pi^{(0,1)}$ and $d_n^{\mathrm{sr}}$, are most strongly constrained by the neutron EDM measurement and again Hg, suggesting that there will be significant correlations between the leptonic and hadronic model parameters in the global analysis. The other diamagnetic systems lead to much weaker limits, but will still be needed to constrain the 3-dimensional hadronic model space. Note that we are fixing the relation between the short-range neutron and proton parameters in the Lagrangian, Eq.(21), but from Tab. 3 we know that different measurements probe different admixtures of these two parameters.

| System $i$ | $d_e\,[e\,\mathrm{cm}]$ | $C_S^{(0)}$ | $C_P^{(0)}$ | $C_T^{(0)}$ |
|---|---|---|---|---|
| Tl | $(7.2 \pm 7.7)\cdot 10^{-28}$ | $(5.9 \pm 6.4)\cdot 10^{-8}$ | $(-2.7 \pm 3.0)\cdot 10^{-6}$ | $(-4.5 \pm 4.9)\cdot 10^{-5}$ |
| Cs | $(-1.4 \pm 5.6)\cdot 10^{-26}$ | $(-2.3 \pm 8.9)\cdot 10^{-6}$ | $(1.3 \pm 5)\cdot 10^{-4}$ | $(-1.1 \pm 4.2)\cdot 10^{-4}$ |
| $^{199}$Hg | $(-1.8 \pm 2.6)\cdot 10^{-28}$ | $(-1.7 \pm 2.5)\cdot 10^{-9}$ | $(3.4 \pm 4.8)\cdot 10^{-8}$ | $(-3.4 \pm 4.9)\cdot 10^{-10}$ |
| $^{129}$Xe | $(2.2 \pm 2.3)\cdot 10^{-25}$ | $(8.3 \pm 8.7)\cdot 10^{-7}$ | $(-1.0 \pm 1.1)\cdot 10^{-5}$ | $(-1.4 \pm 1.5)\cdot 10^{-7}$ |
| $^{171}$Yb | $(5.7 \pm 4.4)\cdot 10^{-25}$ | $(7.5 \pm 5.7)\cdot 10^{-6}$ | $(-1.5 \pm 1.2)\cdot 10^{-4}$ | $(1.6 \pm 1.2)\cdot 10^{-6}$ |
| $^{225}$Ra | $(-7.4 \pm 1.1)\cdot 10^{-23}$ | $(4.7 \pm 7)\cdot 10^{-4}$ | $(-5.7 \pm 8.5)\cdot 10^{-3}$ | $(-8.9 \pm 13)\cdot 10^{-5}$ |
| TlF | $(-2.1 \pm 3.6)\cdot 10^{-25}$ | $(-3.0 \pm 5.1)\cdot 10^{-6}$ | $(-7.1 \pm 12)\cdot 10^{-5}$ | $(-3.6 \pm 6.1)\cdot 10^{-8}$ |
| HfF$^+$ | $(-1.3 \pm 2.1)\cdot 10^{-30}$ | $(-1.4 \pm 2.3)\cdot 10^{-10}$ | | |
| ThO | $(4.3 \pm 4.1)\cdot 10^{-30}$ | $(2.9 \pm 2.7)\cdot 10^{-10}$ | | |
| YbF | $(-2.4 \pm 5.9)\cdot 10^{-28}$ | $(-2.7 \pm 6.6)\cdot 10^{-8}$ | | |
| | $g_\pi^{(0)}$ | $g_\pi^{(1)}$ | $d_n^{\mathrm{sr}}$ | $d_p^{\mathrm{sr}}$ |
| $n$ | $(0 \pm 8.1)\cdot 10^{-13}$ | $(0 \pm 4.1)\cdot 10^{-11}$ | $(0 \pm 1.1)\cdot 10^{-26}$ | $(0 \pm 1.1)\cdot 10^{-26}$ |
| $^{199}$Hg | $(-1.9 \pm 2.7)\cdot 10^{-13}$ | $(1.4 \pm 2.0)\cdot 10^{-13}$ | $(-1.4 \pm 2.0)\cdot 10^{-26}$ | $(-1.4 \pm 2.0)\cdot 10^{-25}$ |
| $^{129}$Xe | $(4.4 \pm 4.6)\cdot 10^{-9}$ | $(8 \pm 8.3)\cdot 10^{-10}$ | $(-1.0 \pm 1.1)\cdot 10^{-23}$ | $(-5.0 \pm 5.2)\cdot 10^{-23}$ |
| $^{171}$Yb | $(7.2 \pm 5.5)\cdot 10^{-10}$ | $(-5.2 \pm 4.0)\cdot 10^{-10}$ | $(6.0 \pm 4.6)\cdot 10^{-23}$ | $(6.0 \pm 4.6)\cdot 10^{-22}$ |
| $^{225}$Ra | $(2.4 \pm 3.5)\cdot 10^{-9}$ | $(-5.8 \pm 8.7)\cdot 10^{-10}$ | $(-7.5 \pm 11)\cdot 10^{-21}$ | $(-3.6 \pm 5.4)\cdot 10^{-20}$ |
| TlF | $(-9.0 \pm 15)\cdot 10^{-10}$ | $(1.1 \pm 1.8)\cdot 10^{-10}$ | $(1.8 \pm 3.1)\cdot 10^{-22}$ | $(3.7 \pm 6.3)\cdot 10^{-23}$ |

Table 4: Single-parameter ranges allowed by each of the EDM measurements given in Tab. 1.

## 4   Global analysis

To combine the available EDM measurements and analyze them in terms of the hadronic-scale Lagrangian and its parameters, Eq.(22), we use the established SFITTER analysis tool. It constructs a global likelihood with a comprehensive uncertainty treatment and analyses it in terms of high-dimensional correlations. Lower-dimensional and one-dimensional likelihoods for the individual model parameters can be derived by profiling or marginalization, depending on the preferred statistical framework. If we assume that experimental uncertainties are Gaussian, profiling and marginalization have to lead to identical results. For the theory uncertainties, discussed in Sec. 4.5, the difference between the two approaches makes a formal, but not significant difference.

### 4.1   SFitter framework

SFITTER [34–36] has been developed for global analyses of LHC measurements in the context of BSM physics [94] and Higgs and top properties [38,95–97], including comprehensive studies of the connection between EFTs and UV-completions of the SM [98,99]. It has a focus on its uncertainty treatment, including theory uncertainties [39], the connection between Bayesian and frequentist approaches [35,37], and published experimental likelihoods [38].

Our first SFITTER analysis of EDMs relates the 11 measurements from Tab. 1 to the seven model parameters in Eq.(22). In general, SFITTER includes statistical, systematic, and theory uncertainties. At the heart of SFITTER is the fully exclusive likelihood as a function of model and nuisance parameters. All measurements are described as uncorrelated, with the individual statistical uncertainties described by Poisson or Gaussian likelihood. Statistical uncertainties are, usually, uncorrelated as well and described by a Poisson distribution, turning into a Gaussian for high statistics. Experimental systematics are assumed to have a Gaussian shape, but

can be described by any nuisance parameter. This Gaussian shape is justified for parameters which are measured elsewhere. We use it for all experimental uncertainties, assuming that the measurements are at least as much dominated by statistical uncertainty, with error distribution appropriate for count-rate limited frequency measurements.

For the EDM analysis the situation is then relatively simple. First, from Eq.(23) we know that all observables depend on the model parameters linearly. Second, we can combine the statistical and systematic experimental uncertainties into the symmetric Gaussian error bars given in Tab. 1. Finally, we do not have to consider nuisance parameters, if we assume that the likelihood has a Gaussian form for each independent measurement. This Gaussian assumption also implies that for uncorrelated uncertainties a profile likelihood and a Bayesian marginalization will give the same result.

Theory uncertainties have no well-defined likelihood shape, and no maximum, but they can be thought of as a range [39]. A flat theory uncertainty is not parametrization-invariant, as one would have expected from a fixed range, but without a preferred central value we consider it conservative. In SFITTER it is implemented through an allowed shift of the prediction at no cost in the likelihoods. For the EDM analysis, theory uncertainties affect the $\alpha$-values with their central values given in Tab. 3.

To construct the exclusive likelihood, SFITTER evaluates EDM predictions over the entire model parameter space. It uses a Markov chain to encode the likelihood in the distribution of points covering the model space. A helpful aspect, common to many BSM analyses, is that we can safely assume the global minimum of the likelihood to be at the SM parameter point. To remove nuisance parameters or to extract limits on a reduced number of model parameters, SFITTER can employ a profile likelihood or a Bayesian marginalization [35, 37]. These two methods give different results, with the exception of uncorrelated Gaussians. Profiling over flat theory uncertainties and Gaussian experimental uncertainties leads to the RFit [100] prescription, profiling over two parameters with flat likelihood leads to linearly added uncertainties even for uncorrelated parameters.

While Eq.(23) suggest a homogeneous set of model parameters, the typical sizes of the model parameters in Eq.(22) and the $\alpha$-values in Tab. 3 can be extremely different. For numerical reasons, we internally re-scale each model parameter and each $\alpha$-value such that all model parameters are evaluated with similar size. Concretely, this means rescaling $d_e$ by a factor $10^{29}$, $C_S^{(0)}$ by $10^9$, $g_\pi^{(1)}$ and $g_\pi^{(0)}$ by $10^{10}$, $C_T^{(0)}$ by $10^8$, $C_P^{(0)}$ by $10^6$, and $d_{n/p}^{\mathrm{sr}}$ by $10^{23}$. These rescalings are also reflected in the way we present our results.

## 4.2 Well-constrained model sub-space

As a starting point of the global analysis and to understand the main features, we consider a sub-space of well-constrained parameters. Following our discussion in Sec. 3.4 we expect the semileptonic parameters $d_e$ and $C_S^{(0)}$ to be constrained well by the paramagnetic molecules HfF$^+$ and ThO. Similarly, the hadronic parameters $g_\pi^{(0)}$ and $d_n^{\mathrm{sr}}$ are strongly constrained by the neutron and Hg EDMs. This means the model sub-space

$$\left\{ d_e, C_S, g_\pi^{(0)}, d_n^{\mathrm{sr}} \right\} \tag{36}$$

should be constrained well by the full set of measurements given in Tab. 1.

The interesting question is how the constraints on these four model parameters are correlated. In Fig. 1 we show these correlations extracted as 2-dimensional profile likelihoods from the fully exclusive, 4-dimensional likelihood. Three structural aspects stick out: (i) a strong

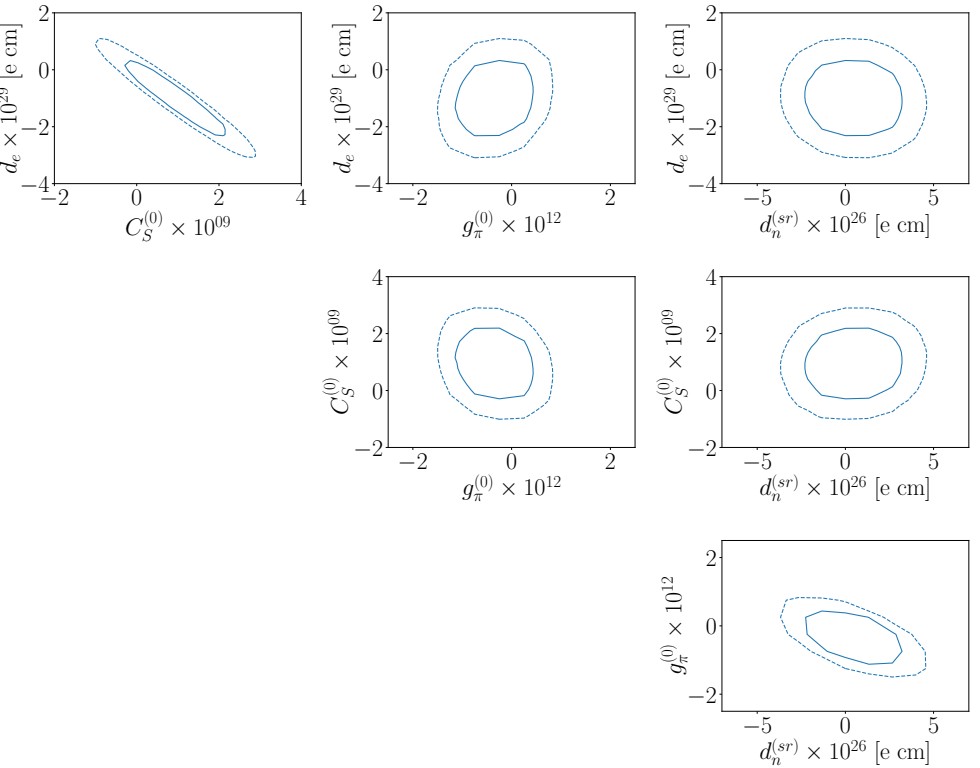

Figure 1: Correlations from the 4-dimensional analysis of $\{d_e, C_S^{(0)}, g_\pi^{(0)}, d_n^{\mathrm{sr}}\}$, based on all EDM measurements but neglecting theory uncertainties. The ellipses indicate 68% and 95% CL.

anti-correlation between $d_e$ and $C_S^{(0)}$; (ii) a strong anti-correlation between $g_\pi^{(0)}$ and $d_n^{\mathrm{sr}}$; and (iii) essentially no correlations between the $\{d_e, C_S^{(0)}\}$ and $\{g_\pi^{(0)}, d_n^{\mathrm{sr}}\}$ parameter subsets.

Within the leptonic sector, the strong correlation between $d_e$ and $C_S^{(0)}$ and its independence from the remaining parameter space is expected to remain for the full global analysis. The reason is that it is induced by the strongest measurements of HfF$^+$ and ThO, and according to our parameterization as shown in Tabs. 3 and 4 those two measurements are not affected by any other model parameter. This means the upper-left panel of Fig. 1 factorizes from our global EDM analysis, and we can consider the remaining model parameters separately and without the HfF$^+$ and ThO measurements.

For the hadronic parameters the situation is different. Again the neutron and Hg measurements are three orders more constraining than the other measurements. However, as the two model parameters we could as well have chosen $g_\pi^{(0)}$ vs $g_\pi^{(1)}$, without any change in the conclusion. This means that we have to expand the hadronic parameter space next, to see what patterns emerge. The parameters $d_e$ and $C_S^{(0)}$ will be kept factorized for the rest of our global analysis.

## 4.3  Hadronic parameters from diamagnetic systems

For the purely hadronic sector, we define a second simplified model parameter space,

$$\left\{ g_\pi^{(0)}, g_\pi^{(1)}, d_n^{\mathrm{sr}} \right\}. \tag{37}$$

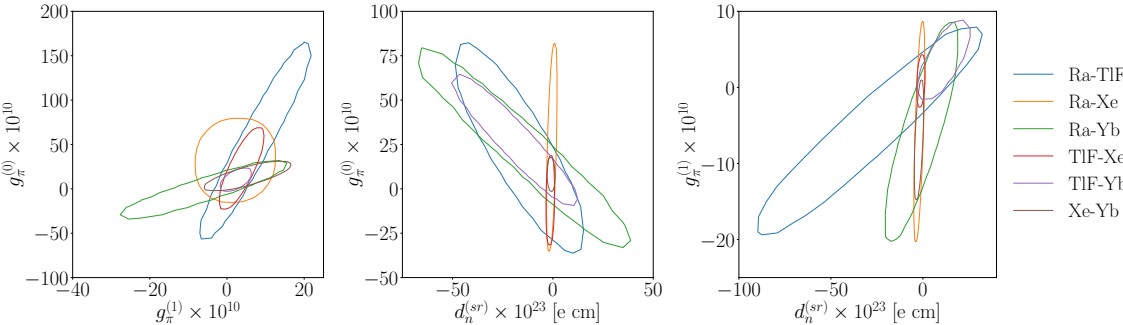

Figure 2: Correlations from three 2-dimensional analyses in the $\{g_\pi^{(0)}, g_\pi^{(1)}, d_n^{\text{sr}}\}$ parameter space, each based on a different pair of diamagnetic EDM measurements, as indicated by the color. The ellipses indicate 68% CL, neglecting theory uncertainties.

All three parameters are constrained by the neutron and diamagnetic EDMs, while we know from the above discussion that the constraints from diamagnetic systems on $d_e$ and $C_S^{(0)}$ are weaker than those from their paramagnetic counterparts.

From Tab. 4 we see that the neutron and and Hg measurements strongly constrain two of the three hadronic model parameters in Eq. (37). To understand the correlations structure of the remaining diamagnetic measurements we show the correlated constraints of the six possible pairs of diamagnetic measurements on the different 2-dimensional sub-spaces of Eq. (37).

Starting with the left panel of Fig. 2, the different pairs of measurements constrain the $g_\pi^{(0)}$ vs. $g_\pi^{(1)}$ plane with different correlation patterns, implying that a global analysis will constrain the 3-dimensional hadronic subspace much better than any single pair of measurements. In the center panel, the situation changes when we look at the correlation with the neutron EDM-parameter $d_n^{\text{sr}}$. Three combinations are aligned to similar negative correlations between $d_n^{\text{sr}}$ and $g_\pi^{(0)}$. In the right panel, the situation is similar for $g_\pi^{(1)}$, with a positive correlation and less striking. In both cases the exception are the combinations of Xe with Ra, TlF, and Yb, which constrain $d_n^{\text{sr}}$ extremely well and without any correlation with $g_\pi^{(0,1)}$. While this sounds like an advantage, we remind ourselves that from Tab. 4 and Fig. 1 we know that those limits on $d_n^{\text{sr}}$ are three orders of magnitude weaker than what can be expected from including the neutron and and Hg measurements.

Altogether, Fig. 2 confirms that the constraints of the sub-leading four diamagnetic measurements on the 3-dimensional hadronic parameter space of Eq. (37) are correlated in a non-trivial manner. Evaluating these correlations requires a global analysis of the, formally, over-constraining set of diamagnetic measurements.

## 4.4 Poorly constrained model parameters

Finally, we combine the effects of all remaining parameters,

$$\left\{ C_T^{(0)}, C_P^{(0)}, g_\pi^{(0)}, g_\pi^{(1)}, d_n^{\text{sr}} \right\}, \tag{38}$$

ignoring the ThO and HfF$^+$ measurements, which constrain the factorized $d_e - C_S^{(0)}$ subspace, and also ignoring the neutron and Hg measurements. The latter constrain the above parameters, but because they are much stronger than all other measurements, they will induce narrow correlations in the allowed 5-dimensional parameter space.

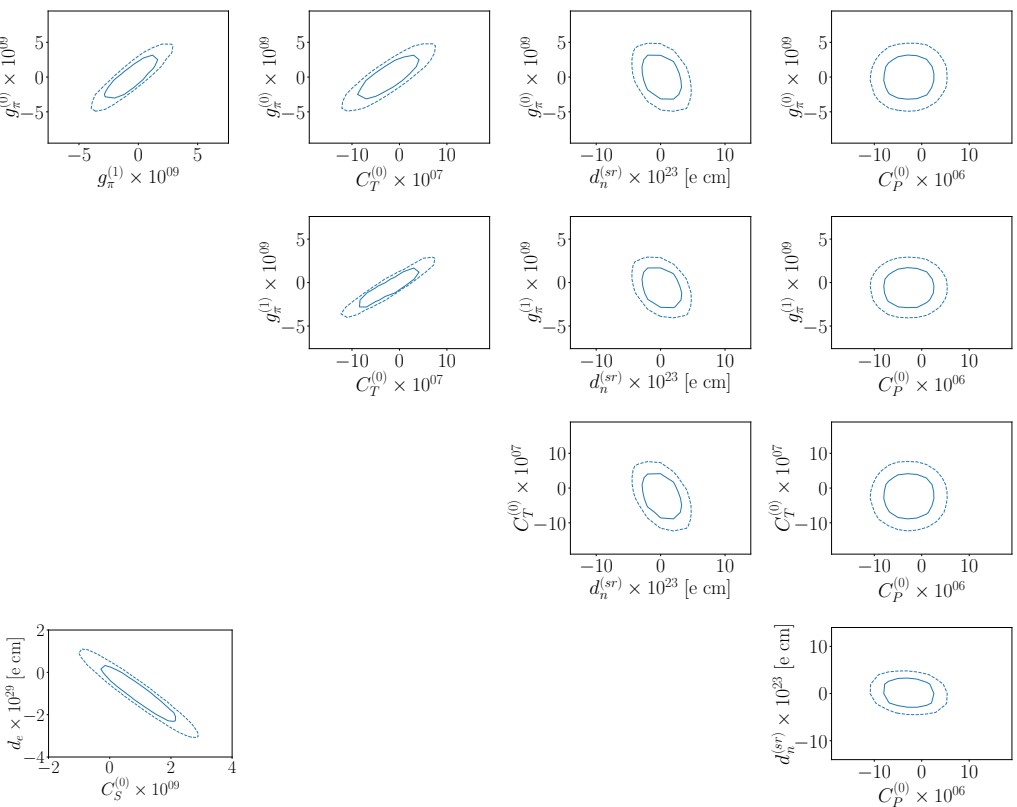

Figure 3: Correlations from the 5-dimensional analysis of $\{C_T^{(0)}, C_P^{(0)}, g_\pi^{(0)}, g_\pi^{(1)}, d_n^{\mathrm{sr}}\}$ and the factorized $d_e - C_S^{(0)}$ plane from Fig. 1. We ignore the neutron and Hg measurements, which induce narrow correlation patterns in the 5-dimensional parameters space and do not affect the profiled 2-dimensional correlations. The ellipses indicate 68% and 95% CL, neglecting theory uncertainties.

Narrow correlations in a higher-dimensional parameter space vanish when we profile the likelihood onto 2-dimensional correlations or even single model parameters. As an example, consider a 2-dimensional parameter space constrained by one strong and one weak measurement. The strong measurement leads to a narrow correlation between the two parameters. When we extract the profile likelihood for one model parameter we can adjust the other model parameter such that the two parameters trace this narrow correlation. This way, the entire length of the correlation pattern gets projected onto the 1-dimensional profile likelihoods. The weak measurement dominates the individual profile likelihoods, while the strong measurement allows us to link the second model parameter from a first model parameter precisely.

In our case this implies that as long as the correlations from the neutron and Hg measurements cross the entire parameter space, we can ignore these two measurements and their correlation patterns in the following discussion of 2-dimensional correlations and single-parameter profile likelihoods. Losing the best few measurements contributing to the global analysis is an unfortunate effect of the conservative profile likelihood approach, but for standard error ellipses it should also exist for Bayesian marginalization. The main difference is that this kind of effect is numerically extremely challenging to compute using marginalization, while it is, essentially, trivial for the profile likelihood.

Given these considerations, we are left with five model parameters, constrained by seven measurements of similar constraining power. We expect correlation patterns in the allowed

parameter space, but no flat directions. Indeed, in Fig. 3 we see that all Lagrangian parameters are nicely constrained. The allowed range, for instance for $g_\pi^{(0)}$ is of the order $10^{-9}$. This can be compared to the constraints from Fig. 1, of the order $10^{-12}$. The same hierarchy of measurements can be observed for $g_\pi^{(1)}$ and for $d_n^{sr}$, as confirmed by Tab. 4. This means that the 5-dimensional allowed parameter space illustrated by Fig. 3 is crossed by two correlation patterns, roughly three orders of magnitude more narrow than the full parameter space. We emphasize that explaining this extremely narrow correlation poses a fine-tuning problem in model parameter space, which the profile likelihood does not address.

## 4.5   Theory uncertainties

Theory uncertainties always appear when we use quantum field theory to predict observables, like EDMs, from Lagrangian parameters: no calculation method is arbitrarily precise, and a variety of systematic errors can effect accuracy. While there is some hope that we can estimate and control uncertainties for small expansion parameters, the uncertainties associated with QCD observables at low energies (whether from lattice calculations or sum-rule estimates) are much more tricky to estimate. The precision of nuclear physics calculations, and their links to effective quantum field theory, are also challenging to quantify. On the other hand we have to estimate all of these uncertainties, and we can only ignore them after having shown that they are significantly smaller than the experimental uncertainties of the associated measurements.

For our global EDM analysis, theory uncertainties affect the $\alpha_{i,c_j}$ in Eq.(23). The estimated

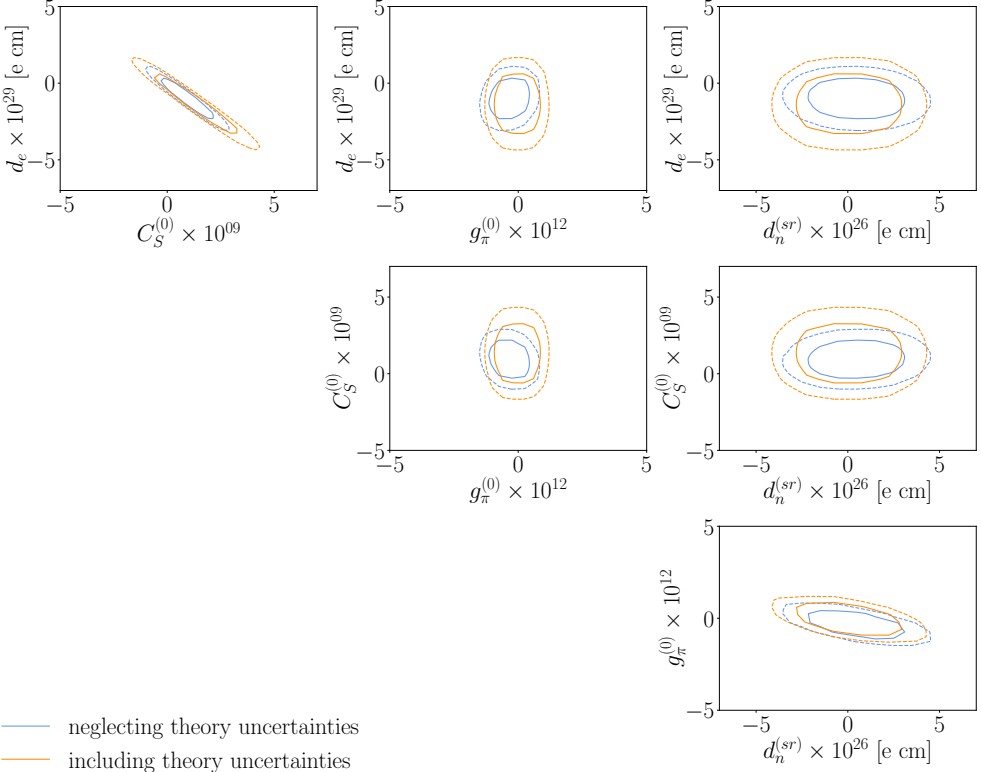

Figure 4: Correlations from the 4-dimensional analysis of $\{d_e, C_S^{(0)}, g_\pi^{(0)}, d_n^{sr}\}$. The orange curves show the effect of theory uncertainties on the results of Fig. 1. The ellipses indicate 68% and 95% CL.

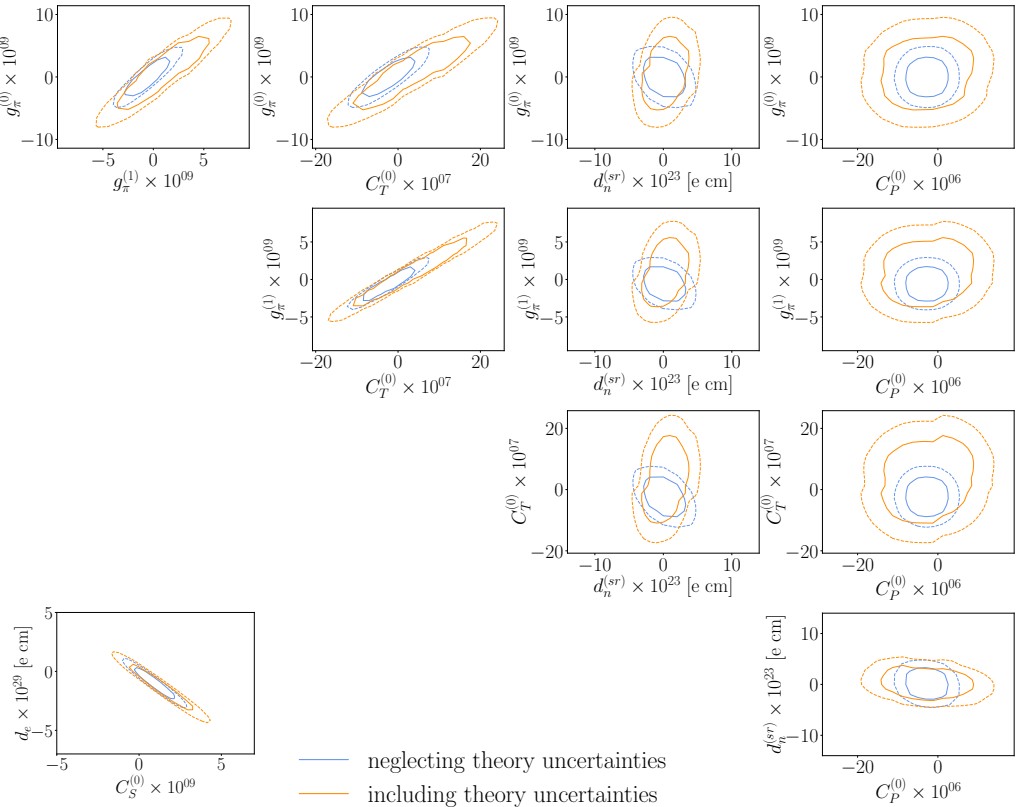

Figure 5: Correlations from the 5-dimensional analysis of $\{C_T^{(0)}, C_P^{(0)}, g_\pi^{(0)}, g_\pi^{(1)}, d_n^{\text{sr}}\}$, and the factorized $d_e - C_S^{(0)}$ plane from Fig. 4. The orange curves show the effect of theory uncertainties on the results of Fig 3. The ellipses indicate 68% and 95% CL.

allowable ranges of each coefficient are given in Tab. 3. We include them in SFITTER as uncorrelated theory uncertainties, an assumption that can be modified if necessary. Because of the flat likelihood as a function of the theory nuisance parameter, the profile likelihood approach leads to the theory uncertainties adding linearly, weighted by the respective model parameter. Profiling over independent $\alpha$-ranges simplifies the numerical evaluation in two ways: first, any parameter–observable pair for which $\alpha$ is compatible with zero will effectively be removed from the global analysis, because the optimal choice of $\alpha$ will remove all contributions from the corresponding model parameter; second, even if we cannot choose $\alpha$ such that measurement and prediction agree, we can choose it to maximize the likelihood and to minimize the impact of the measurement, which means we choose the smallest allowed absolute value of $\alpha$.

As in Sec. 4.2 we start with the well-constrained 4-dimensional subspace $\{d_e, C_S^{(0)}, g_\pi^{(0)}, d_n^{\text{sr}}\}$. Again, $d_e$ and $C_S^{(0)}$ are constrained by the paramagnetic molecules HfF$^+$ and ThO, just as without theory uncertainties. From Tab. 3 we see that we can ignore the theory uncertainty in relating the electron EDM parameter $d_e$ to these systems. In the hadronic sector, the theory uncertainties affecting the relation of $g_\pi^{(0)}$ and $d_n^{\text{sr}}$ to the neutron and Hg EDMs is reasonably small, although for the neutron this situation should not be over-interpreted. The quoted uncertainty arises from propagating ranges for the constants within Eq.(26), and not from careful evaluations of the chiral expansion itself. In Fig. 4 we show the numerical impact of the theory uncertainties on the 2-dimensional correlations. The slightly stronger HfF$^+$ measurement, which determines the width of the correlation pattern, is only minimally affected by the theory uncertainties, while the larger theory uncertainties on the ThO measurement extend the

length of the ellipse visibly.

Moving on, we can now look at the effect of the theory uncertainties on the less-constrained hadronic sector for diamagnetic systems, discussed in Sec. 4.3. Here, Tab. 3 shows sizeable, order-one theory uncertainties. In addition, some of the $\alpha$-values include an allowed zero value when we include theory uncertainties. Specifically, $g_\pi^{(0)}$ will no longer be constrained by the Xe measurement and $g_\pi^{(1)}$ will lose the Hg constraint. In addition, some of the theory uncertainties in the diamagnetic sector are large, so we expect sizeable impact.

As for the case without theory uncertainties, the neutron and Hg limits are much more constraining than the other measurements of the hadronic sector. Following the argument given in Sec. 4.4, this means that 2-dimensional correlations and single-parameter limits extracted by profiling the likelihoods will not be impacted by these strong measurements. The shift in the constrained 2-dimensional correlations are shown in Fig. 5. In comparison their effect on the factorized parameters $d_e$ and $C_S^{(0)}$ (copied from Fig. 4) is mild, and the constraints on the hadronic model parameters are consistently weaker. The only exception is $d_n^{\text{sr}}$, since as we know from Sec. 4.3 the measurements are hardly correlated and the theory uncertainties are smaller than order-one.

# 5 Outlook

EDMs are extremely sensitive, targeted probes of one of the most important symmetries of elementary particles, directly related to the observed baryon asymmetry in the Universe. The number of EDM measurements in very different systems has grown rapidly in recent years, leading to the question of how the different measurements can contribute to constraining and understanding CP violation in terms of a fundamental Lagrangian.

We can choose different Lagrangians to answer this question, starting with UV-complete models versus EFTs. In the absence of a specific hint for BSM physics we choose an EFT description. Next, we have a choice of different energy scales with different degrees of freedom. For our first SFITTER analysis we rely on the hadronic-scale Lagrangian, valid at the GeV scale and describing the interactions of electrons and nucleons. After relating the hadronic-scale Lagrangian to its weak-scale SMEFT counterpart we want to constrain the seven Lagrangian parameters given in Fig. 6 through 11 independent measurements given in Tab. 1.

As a toy analysis we look at single-parameter constraints from individual EDM measurements. These limits all driven by the same small set of highly constraining measurements, like the paramagnetic molecular ion HfF$^+$, the neutron EDM, or the diamagnetic atom Hg. The extremely strong constraints indicated in Fig. 6 do not allow for a cancellation of contributions from two model parameters to a given measurement, at the price of creating a fine tuning problem. While the single-parameter estimates indicate the strength of an experiment looking for a sign of CP violation, they should not be confused with a measurement of a given parameter.

For our global analysis we use SFITTER, with its focus on the statistical interpretation and a comprehensive uncertainty treatment. First, we ignore all theory uncertainties and only consider experimental uncertainties as uncorrelated and Gaussian (combining the statistical and systematic uncertainties reported in the respective papers). In this case Bayesian marginalization and profile likelihood give the same result.

In Sec. 4.2 we find that a small set of powerful measurements constrains the electron-hadron interactions as well as a subset of the hadronic sector. The correlated limits on the

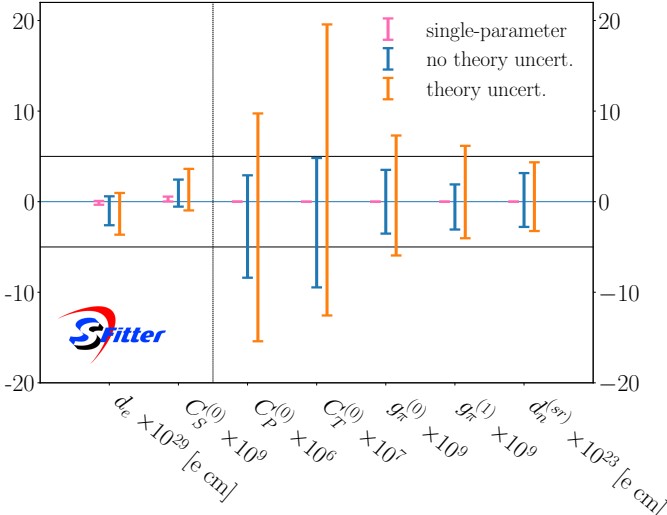

Figure 6: Constraints from the global EDM analysis on the parameters of the hadronic-scale Lagrangian. We show (i) hugely over-constrained single-parameter ranges allowed by the best available measurement; (ii) over-optimistic allowed ranges for profiled single parameters, ignoring theory uncertainties; (iii) allowed ranges for profiled single parameters including experimental and theory uncertainties.

electron EDM parameter $d_e$ and the scalar coupling $C_S^{(0)}$ are especially strong and factorize from the hadronic sector. Next, we find in Sec. 4.3 that the constraints on the hadronic sector from the diamagnetic systems show rich correlations, motivating our global analysis described in Sec. 4.4. For the hadronic parameters, the narrow correlations from the strong neutron and Hg constraints do not appear in profiled 2-dimensional correlations or single-parameter limits. As a result, the hadronic model parameters are constrained much worse than the single-parameter results suggest, as can be in Fig. 6.

Finally, we show the same limits including theory uncertainties. Such theory uncertainties always appear when we relate measurements to fundamental Lagrangian parameters. We assume a flat likelihood within allowed ranges of the factors relating the Lagrangian parameters to the EDM predictions. While the impact of the theory uncertainties on the factorized semileptonic sector is relatively mild, the correlated analysis of the hadronic parameters leads to a further weakening of the constraints on the individual model parameters.

## Acknowledgments

We gratefully acknowledge extremely helpful discussions and correspondence with Tim Chupp, Vincenzo Cirigliano, Jordy de Vries, Victor Flambaum, Timo Fleig, Maxim Pospelov, and Adam Ritz. We also thank Duarte Azevedo for his contributions at an early phase of this project. We would like to thank the ECT* Trento for the hospitality while submitting this work and presenting the results for the time.

NE is funded by the Heidelberg IMPRS *Precision Tests of Fundamental Symmetries*. This research is supported by the Deutsche Forschungsgemeinschaft (DFG, German Research Foundation) under grant 396021762 – TRR 257: *Particle Physics Phenomenology after the Higgs Discovery* and through Germany's Excellence Strategy EXC 2181/1 – 390900948 (the *Heidelberg STRUCTURES Excellence Cluster*). Furthermore, we acknowledge support by the state

of Baden-Württemberg through bwHPC and the German Research Foundation (DFG) through grant no INST 39/963-1 FUGG (bwForCluster NEMO).

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
