# Peer review of "A Global View of the EDM Landscape"

_SciPost Physics_

## Round 2 · Referee Report · Guillaume Pignol (Referee 1) · 2025-6-7

Report

This article presents a comprehensive global analysis of all electric dipole moment (EDM) measurements, framed within a hadronic-scale effective field theory and implemented using the SFitter tool. The main achievement of the paper is to provide a coherent methodology for translating EDM constraints into limits on fundamental CP-violating (CPV) parameters, accounting for theoretical uncertainties and correlations. The work is significant, well executed, and provides a useful tool to assess the global landscape of EDM constraints. The strategy —to treat all EDM measurements as part of a coherent program to probe fundamental sources of CP violation, without imposing single-source assumptions— is timely and appropriate.

By organizing existing and prospective EDM measurements into a unified, model-independent framework, the authors offer a synergetic link between (i) the diverse low-energy precision measurements, (ii) the ongoing theoretical program to connect observables to fundamental CPV parameters, (iii) advanced techniques of global fits in the context of effective field theory analyses. The work opens a pathway for follow-up efforts, notably in interpreting new experimental results and refining theoretical inputs. The analysis improves on earlier EDM global fits (Chupp and Ramsey-Musolf, 2015), both by updating the experimental and theoretical landscape and by providing a concrete implementation that can be adapted and extended. The treatment of uncertainties, including the impact of theory errors on global constraints, is particularly welcome and necessary for interpreting the EDM searches in any concrete BSM model.

The paper meets the general acceptance criteria of SciPost Physics: * It is clearly written, with an intelligible presentation of the goals, methods, and conclusions. * The technical content is detailed enough to be reproduced, with full references to the necessary inputs. * The citations are generally appropriate and representative of the relevant literature. * The abstract and introduction provide sufficient context and explain the significance of the work. * The conclusion is appropriately balanced, outlining both the reach and the limitations of the analysis, and offering realistic perspectives.

I recommend publication after minor revisions, as outlined in the separate list of comments. These are mostly clarifications and small corrections that will improve the readability and precision of the manuscript, without affecting its core scientific content.

Requested changes

1- Before Eq. (10): 4-fermion operators The paragraph gives the impression that the 4-fermion operators in the SMEFT generate only the semileptonic couplings in Eq. (10). But 4-quark contact operators also generate nucleon EDMs and other hadronic TV couplings in Eq. (7).

2- Indexing in Eq. (29) The sentence "The superscript (m) indicates the molecular systems." is confusing. Isn’t "i" the index that labels the system? Please clarify the notation.

3- What are semileptonic parameters? The phrase "we are not aware of any established values for the coefficients of semileptonic or hadronic parameters" is unclear. Isn’t C_S a semileptonic parameter?

4- Wrong table reference in "At present, as can be seen from Tab.s 2 and 3..." The claim that opposite signs of alpha_i,de and alpha_i,CS can be seen in Tables 2 and 3 is incorrect – this information appears only in Table 3.

5- Explanation of Eq. (35) too compact Understanding Eq. (35) is not straightforward unless one is already familiar with the relevant literature. Please expand the explanation. What is the "system-specific" coefficient k_{i,S}? Clarify what is meant by "system" here. Consider writing separate expressions for the atomic EDM: one term from the Schiff moment, one from semileptonic interactions, etc. For example: d_A = k_i * S_i + c_i * M_i + alpha * C_T, then S_i = ...

6- Misleading language on Schiff moments The sentence "The Schiff moment itself can, in principle, be large." is misleading. The Schiff moment vanishes in the absence of CP violation. What can be large is the coefficient relating the CPV coupling (e.g., g_0) to the Schiff moment, due to nuclear enhancement.

7- Possible typo in Table 5 for 199Hg The value a_0 = 0.01 (+0.4 / –0.005) seems wrong. The upper error should likely be +0.04, not +0.4.

8- CPV in mercury: theory error not small The sentence stating that "theory uncertainties are reasonably small" for g_pi^(0) and d_n in relation to the neutron and Hg EDMs is questionable. For mercury, theoretical uncertainties significantly degrade the bound.

9- Single-parameter ranges and theory errors The single-parameter limits in Section 3.4 do not include theoretical uncertainties, but this is not stated explicitly in the section. The reader has to infer it indirectly after reading the rest of the paper.

10- Improve Figure 6 Figure 6 is the central ("money") plot of the paper. It deserves better presentation. In particular: are the horizontal and vertical dashed lines intentional?

Recommendation

Ask for minor revision

---

## Round 2 · Referee Report · Jaco ter Hoeve (Referee 2) · 2025-6-18

Strengths

  1. Innovative
  2. Relevant
  3. Well structured

Weaknesses

  1. Fitting framework can be explained more explicitly
  2. Limited outlook and/or future directions
  3. No renormalization group evolution yet

Report

The manuscript performs a global analysis of a wide range of EDM measurements using the SFitter framework with a careful treatment of statistical, systematic and theoretical uncertainties. A global interpretation in terms of the model parameters describing the EDM is especially encouraging as it properly accounts for correlations, and moreover, facilitates the addition of (new) EDM measurements and beyond in a single unified framework. The authors also show the crucial role played by theory uncertainties by demonstrating how constraints weaken upon including them. The manuscript is clearly written and well presented. All in all, I recommend this work for publication after minor revisions, please see the requested changes below.

Requested changes

  1. At the moment, it is unclear to me whether or not renormalisation group effects have been accounted for. In the introduction to Sect. 2, it is explicitly mentioned that they are not, while the statement "When we evolve our EFT to the experimentally relevant GeV scale" in beginning of Sect. 2.2 seems to suggest they are. Please clarify this.

  2. The introduction refers to challenges in global analyses without mentioning what they really are. This should be spelled out in more detail (it will also further motivate the current work!). By the way, "all global analyses" suggests more than one reference, while only [35] is given. Please either give more references, or refine the statement.

  3. In Eq. 1, the dipole and Weinberg contributions are of mass dimensions 5 and 6 respectively, which are already valid EFT operators. Yet, a separate EFT contribution is written on the same line. This might be confusing to some readers.

  4. Between Eq. 12 and 13 it should be stated that the matching is done at tree-level (inspecting Eq. 13, I suppose it is)

  5. Section 4 can benefit from an explicit expression of the likelihood. At the moment, it is described in words, but adding it would make the paper more much more self contained and a bit easier to follow. What is the likelihood that is optimised? How do you optimise?

  6. On page 18, I found the comment "A helpful aspect, common to many BSM analyses, is that that we can safely assume the global minimum of the likelihood to be at the SM parameter point" somewhat confusing. First, referring to likelihoods this should probably be the maximum, and second, there is no reason the best fit point should coincide with the SM. How should I interpret this statement?

  7. Although theory uncertainties are included, Sect. 4.5 should state more clearly where they come from in the case of the current analysis. Please show explicitly how the likelihood is modified in the presence of theory uncertainties, similar to point 5 above. This can be made clearer.

  8. Fig 2. do the elongated ellipses really close or do they correspond to genuine flat directions?

  9. Figs 4. and 5. Please indicate the best fit point and consider adding a table of bounds to the appendix for easy comparison in future studies. Can you explain the non Gaussian behaviour when including theory uncertainties (my understanding is this originates from a flat contribution to the likelihood). Is this correct?

  10. The outlook does not mention many ideas for future directions. How can this work be extended? What can be improved upon?

  11. Figure 6 should be moved to Sect.5 as it deserves more attention. What are the horizontal lines at +/- 5 and the vertical line?

Recommendation

Ask for minor revision

---

## Round 2 · Referee Report · Anonymous (Referee 3) · 2025-7-1

Report

This paper provides a comprehensive global analysis of EDM constraints on new CP-odd physics using the SFitter framework, which for the first time allows for a systematic treatment of correlated theoretical uncertainties. Thus, it brings the correlated analysis of EDM constraints to a new level of rigour, particularly in the analysis of observables with substantial hadronic EDM contributions. This work is topical as the rapid development of molecular EDM experiments presents opportunities for novel probes of hadronic CP violation in the near future.

The paper uses the standard EFT structure of CP-odd operators, is well-organized and well-written, and will be a valuable addition to the literature. I recommend publication after the authors have addressed the minor points raised below.

Requested changes

  1. The choice in eq(21) (or an alternative in eq(44)) to set d_n = - d_p to reduce the number of free parameters would benefit from references to justify its physical motivation. In practice, any such constraint will be correlated with the underlying UV sources of CP-violation, which perhaps should be emphasized.

  2. The substantial theory uncertainty in relating pion-nucleon couplings to the Schiff moment of Hg remains a long-standing barrier to de-correlating various hadronic CP-odd contributions. This also impacts the inferred constraint on d_p, where there is no direct measurement. It may be worth noting, e.g. in section 4.5, how progress in this area (from ab initio methods or others) would impact the global fit in the future.

Recommendation

Ask for minor revision

---

## Editorial Decision

awaiting_resubmission